# A tailored tetravalent peptide displays dual functions to inhibit amyloid β production and aggregation

Waka Sato[1], Miho Watanabe-Takahashi [1], Takuya Murata[2], Naoko Utsunomiya-Tate[2], Jun Motoyama[3], Masataka Anzai[1], Seiko Ishihara[4], Nanako Nishioka[4], Hina Uchiyama[4], Juri Togashi[4], Saeka Nishihara[5], Kiyoshi Kawasaki[5], Takashi Saito [6], Takaomi C. Saido [7], Satoru Funamoto [4✉] & Kiyotaka Nishikawa [1✉]

Inhibition of amyloid-β peptide (Aβ) accumulation in the brain is a promising approach for treatment of Alzheimer's disease (AD). Aβ is produced by β-secretase and γ-secretase in endosomes via sequential proteolysis of amyloid precursor protein (APP). Aβ and APP have a common feature to readily cluster to form multimers. Here, using multivalent peptide library screens, we identified a tetravalent peptide, LME-tet, which binds APP and Aβ via multivalent interactions. In cells, LME-tet-bound APP in the plasma membrane is transported to endosomes, blocking Aβ production through specific inhibition of β-cleavage, but not γ-cleavage. LME-tet further suppresses Aβ aggregation by blocking formation of the β-sheet conformation. Inhibitory effects are not observed with a monomeric peptide, emphasizing the significance of multivalent interactions for mediating these activities. Critically, LME-tet efficiently reduces Aβ levels in the brain of AD model mice, suggesting it may hold promise for treatment of AD.

[1] Department of Molecular Life Sciences, Graduate School of Life and Medical Sciences, Doshisha University, Kyoto, Japan. [2] Faculty of Pharma-Science, Teikyo University, Tokyo, Japan. [3] Laboratory of Developmental Neurobiology, Graduate School of Brain Science, Doshisha University, Kyoto, Japan. [4] Department of Neuropathology, Graduate School of Life and Medical Sciences, Doshisha University, Kyoto, Japan. [5] Faculty of Pharmaceutical Sciences, Doshisha Women's College of Liberal Arts, Kyoto, Japan. [6] Department of Neurocognitive Science, Institute of Brain Science, Graduate School of Medical Sciences, Nagoya City University, Aichi, Japan. [7] Laboratory for Proteolytic Neuroscience, Riken Center For Brain Science, Saitama, Japan. ✉email: sfunamot@mail.doshisha.ac.jp; knishika@mail.doshisha.ac.jp

Alzheimer's disease (AD) is a progressive neurodegenerative disorder in humans, characterized by the presence of senile plaques within the brains of affected individuals, the main constituent of which is amyloid-β peptide (Aβ)[1,2]. In neuronal cells, Aβ is produced by the sequential proteolytic cleavage of amyloid precursor protein (APP), a type I transmembrane protein[3,4]. APP present in the plasma membrane is internalized into early endosomes and then into late/recycling endosomes, where it is cleaved by β-secretase to produce a soluble N-terminal fragment, sAPPβ, and a membrane-bound C-terminal fragment, CTFβ (C99)[5]. C99 is subsequently cleaved by γ-secretase to produce an APP intracellular domain and Aβ[6,7]. Alternatively, within the plasma membrane, APP can be cleaved by α-secretase to produce a soluble N-terminal fragment, sAPPα, and a membrane-bound C-terminal fragment, CTFα[8,9]. Aβ produced in cells is released into the extracellular space, where it aggregates, leading to the formation of amyloid plaques in the brain, the typical pathological feature of AD. Thus, compounds that inhibit production of intracellular Aβ and/or the aggregation of extracellular Aβ in the brain may be effective therapeutic agents against AD.

Inhibitors against β-secretase or γ-secretase are one such class of compounds that have been proposed as a possible approach for suppressing Aβ production. However, inhibition of these enzymes can lead to cleavage defects in a variety of other substrate proteins, resulting in severe side-effects[10,11]. β-secretase-deficient mice, in particular, exhibit a wide range of pathological symptoms, including axon guidance defects[12] and seizures[13], and even conditional β-secretase knock-out in adulthood impairs axon guidance[14]. These observations underscore the need for new therapeutic strategies based on substrate-specific inhibition. Compounds that bind to or modulate APP and as a result, specifically inhibit its β-cleavage, are particularly attractive, as this is the rate-limiting step for Aβ production[15]. However, inhibitors with this substrate-specific activity have not been developed.

Aβ is prone to aggregate into β-sheet conformations, which are detected in the brains of patients with AD, first in the form of Aβ oligomers, and then as Aβ fibrils[16]. Notably, of the various Aβ species processed by γ-secretase, Aβ42 has been shown to have a greater propensity to aggregate compared to Aβ40, the major processed form[17,18]. APP and C99 can also easily oligomerize to form homodimers through their extracellular domains and possibly through their juxtamembrane and transmembrane domains, as well[19–21]. Thus, a compound that tightly binds to APP, C99, and/or Aβ, particularly Aβ42, could represent an ideal therapeutic agent to inhibit production of Aβ by modulating APP metabolism in cells and/or blocking the subsequent extracellular oligomerization of Aβ.

Previously, we developed a method to screen multivalent peptide libraries and identify tetravalent peptides with the ability to directly bind bioactive molecules that function in multi-subunit forms[22], such as Shiga toxin[23–26], a major virulence factor of enterohemorrhagic Escherichia coli, and influenza virus hemagglutinin[27]. Critically, we found that these tetravalent peptides, but not monomeric peptides with the same motifs, bind to their target molecules with high affinity and potently inhibit their pathogenic function. Thus, our findings clearly demonstrate the importance of multivalent interactions for inhibitors that effectively target molecules that form multi-subunit or oligomeric structures. This further suggests that an optimal inhibitor for APP, C99, and/or Aβ should also engage in multivalent interactions with its target(s).

In this study, by targeting the N-terminal region of Aβ (Aβ1–28), we developed a new method for identifying tetravalent peptides that bind this region via multivalent interactions. Using this method, which is based on an affinity-maturation screening of tetravalent peptide libraries synthesized on a cellulose membrane, we identified a series of peptides that tightly bind to Aβ1–28. We further show that one of these peptides, LME-tet, can bind to APP in cells and specifically inhibit the intracellular production of Aβ by blocking its β-cleavage and the subsequent extracellular release of Aβ. In addition, LME-tet can inhibit aggregation of Aβ42 with remarkable potency through direct binding to the Aβ42 and shows protective effects against Aβ accumulation within the brain in a mouse model of AD. Thus, LME-tet may represent a promising therapeutic candidate for treatment of AD in humans suffering from this devastating disease.

## Results

**Affinity-driven screening of tetravalent peptide libraries identifies a tetravalent peptide that efficiently inhibits production of Aβ.** To identify a compound that modulates APP metabolism through direct interaction with APP, C99, and/or Aβ and inhibits production of Aβ in cells, we targeted the N-terminal region of Aβ (Aβ1–28), which is conserved in all three proteins (Fig. 1a). We then developed a new method to identify tetravalent peptides that bind Aβ1–28 with high affinity through multivalent interactions, which is based on affinity-driven screening of tetravalent peptide libraries synthesized on a cellulose membrane (Fig. 1b). The tetravalent peptide library for the first screen comprised tetravalent peptides with the sequence X-X-X-X-X-X-X-U- (where X is a mixture of all amino acids except Cys, and U is amino hexanoic acid as a spacer), in which each X was replaced individually by a fixed amino acid, excluding Cys (Supplementary Fig. 1a). The membrane containing the peptide library was blotted with biotinylated Aβ1–28, and the amount of Aβ1–28 bound to each tetravalent peptide was quantified. Based on results from this initial screen, the second library was designed with a fixed Lys residue at position 6 (Fig. 1c). After five more steps of affinity-based screens, the binding motifs in the seventh libraries were matured, as shown in Fig. 1c and Supplementary Fig. 1b–j. In some steps, two peptide libraries were designed and screened in parallel as separate groups to verify the diversity of the binding motifs (Fig. 1c). Based on results obtained from the seventh libraries (Groups Ic, Id, Ie, If, IIa, and IIb), 19 binding motifs were determined (Fig. 1d), and their tetravalent forms were synthesized as candidate compounds (Fig. 2a).

We then measured the inhibitory effects of these tetravalent peptides on production of intracellular C99 and Aβ and extracellular Aβ in 7WD10 Chinse Hamster Ovary (CHO) cells stably expressing human APP[28]. Among all peptides tested, only LME-tet (containing the Pro-Lys-Leu-Arg-Met-Lys-Glu motif) was able to reduce the amount of extracellular Aβ to 59% of that in the control, although the intracellular C99 and Aβ levels were not significantly affected (Fig. 2b). The total amount of extracellular Aβ was 30.2 ± 4.4 (mean ± SE; $n = 4$) times higher than that of intracellular Aβ after incubation of 7WD10 cells for 48 h (Supplementary Fig. 2), indicating that the decrease in the amount of extracellular Aβ caused by LME-tet treatment substantially reflects the inhibitory effect of LME-tet on Aβ production in cells. In contrast, we did not detect any inhibitory effects on production of Aβ with QME-tet, NME-tet, SME-tet, PME-tet, and GME-tet, each of which have one amino acid difference at position 3 compared to LME-tet, highlighting the importance of Leu at this position. Time-course assays further show that LME-tet inhibits the production of Aβ in a dose-dependent manner (Fig. 2c, left panel), whereas a monomeric peptide with the same motif, Pro-Lys-Leu-Arg-Met-Lys-Glu (referred to as LME-mono) has no effect on Aβ production (Fig. 2c, right panels). These findings suggest that LME-tet functions to inhibit Aβ production in cells through a multivalent interaction.

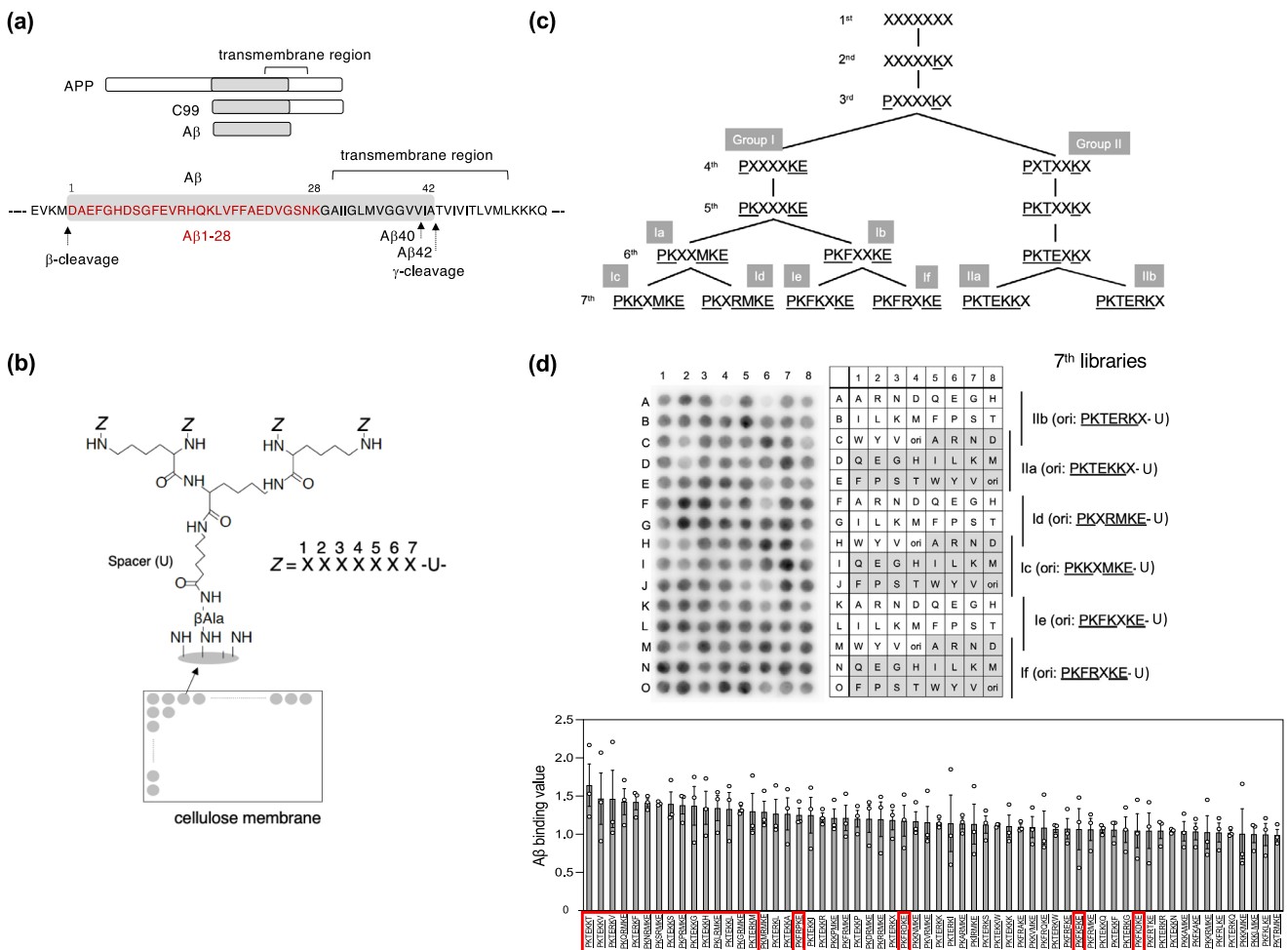

**Fig. 1 Affinity-driven screening of tetravalent peptide libraries synthesized on a membrane identified a series of Aβ1–28 binding peptides. a** Schematic diagram of amyloid precursor protein (APP), C-terminal fragment β (C99), and amyloid-β peptide (Aβ). **b** Schematic showing the structure of our tetravalent peptide library synthesized on a cellulose membrane. The library for the first round of screening contained tetravalent peptides with the sequence X-X-X-X-X-X-X-U- (where X is a mixture of all amino acids except Cys, and U is amino hexanoic acid as a spacer), and each X was replaced individually by a fixed amino acid, excluding Cys (Supplementary Fig. 1a). Membranes were blotted with biotinylated Aβ1–28 (10 μg/ml), and binding of Aβ1–28 to each tetravalent peptide was analyzed, as described in detail in the Methods and in Supplementary Fig. 1a. **c** Schematic outline of the affinity-based maturation of Aβ1–28-binding motifs. The second library was designed based on the motif obtained from the first screen. Similarly, five additional rounds of affinity-based screening were performed to mature the binding motifs and generate the seventh libraries (Supplementary Fig. 1b–j). **d** The library for the seventh round of screening contained peptides with following sequences: P-K-K-X-M-K-E-U- (7th Ic), P-K-X-R-M-K-E-U- (7th Id), P-K-F-K-X-K-E-U- (7th Ie), P-K-F-R-X-K-E-U- (7th If), P-K-T-E-K-K-X-U- (7th IIa), or P-K-T-E-R-K-X-U- (7th IIb), where each X was replaced individually by a fixed amino acid, other than Cys; "ori" indicates the original motif. The binding of Aβ1–28 to each tetravalent peptide was analyzed, as described above, and each screen was performed three times; a representative blotted membrane is shown (upper panel). Mean Aβ-binding values for the top 60 motifs, ± the standard error (SE), are shown in the lower panel (*n* = 3). The top 15 motifs were selected as candidates, and four additional motifs were selected from groups Ie and If, as no motifs from these groups were represented in the top 15; selected motifs are boxed in red.

To further investigate the significance of multivalent interaction between LME-tet and its putative target molecules, the direct binding of LME-tet to Aβ40, Aβ42, and Aβ1–28 was measured using enzyme-linked immunosorbent assays (ELISAs). We found that LME-tet, but not LME-mono, efficiently binds to Aβ1–28, Aβ40 (apparent Kd = 2.7 μM), and Aβ42 (apparent Kd = 0.20 μM) (Fig. 2d). Notably, these values indicate that LME-tet functions based on the clustering effect, a phenomenon by which multivalent interactions between two molecules with multisubunit or oligomeric structure markedly increase the binding affinity as compared to a 1:1 monomeric interaction[23,27,29]. The relatively higher binding affinity to Aβ42 vs. Aβ40 may thus reflect its propensity to aggregate, thus exaggerating the clustering effect.

**LME-tet forms a complex with APP and co-localizes with APP in endosomes**. To determine whether LME-tet binds directly to APP, C99, and/or Aβ in cells, we performed coprecipitation assays. To this end, 7WD10 cells were treated with biotinylated LME-tet or LME-mono for 30 min at 37 °C. Cells were then lysed, and proteins that coprecipitated with each peptide were analyzed by western blot. Results show that APP, C99, and Aβ coprecipitated with LME-tet, but not LME-mono (Fig. 3a), indicating that LME-tet can bind to all three proteins through multivalent interactions in cells. We then assessed whether LME-tet initially targets APP present in the plasma membrane by treating 7WD10 cells with biotinylated LME-tet for 30 min at 4 °C and then performing similar coprecipitation assays. Under these conditions, which inhibit endocytosis, APP was efficiently coprecipitated

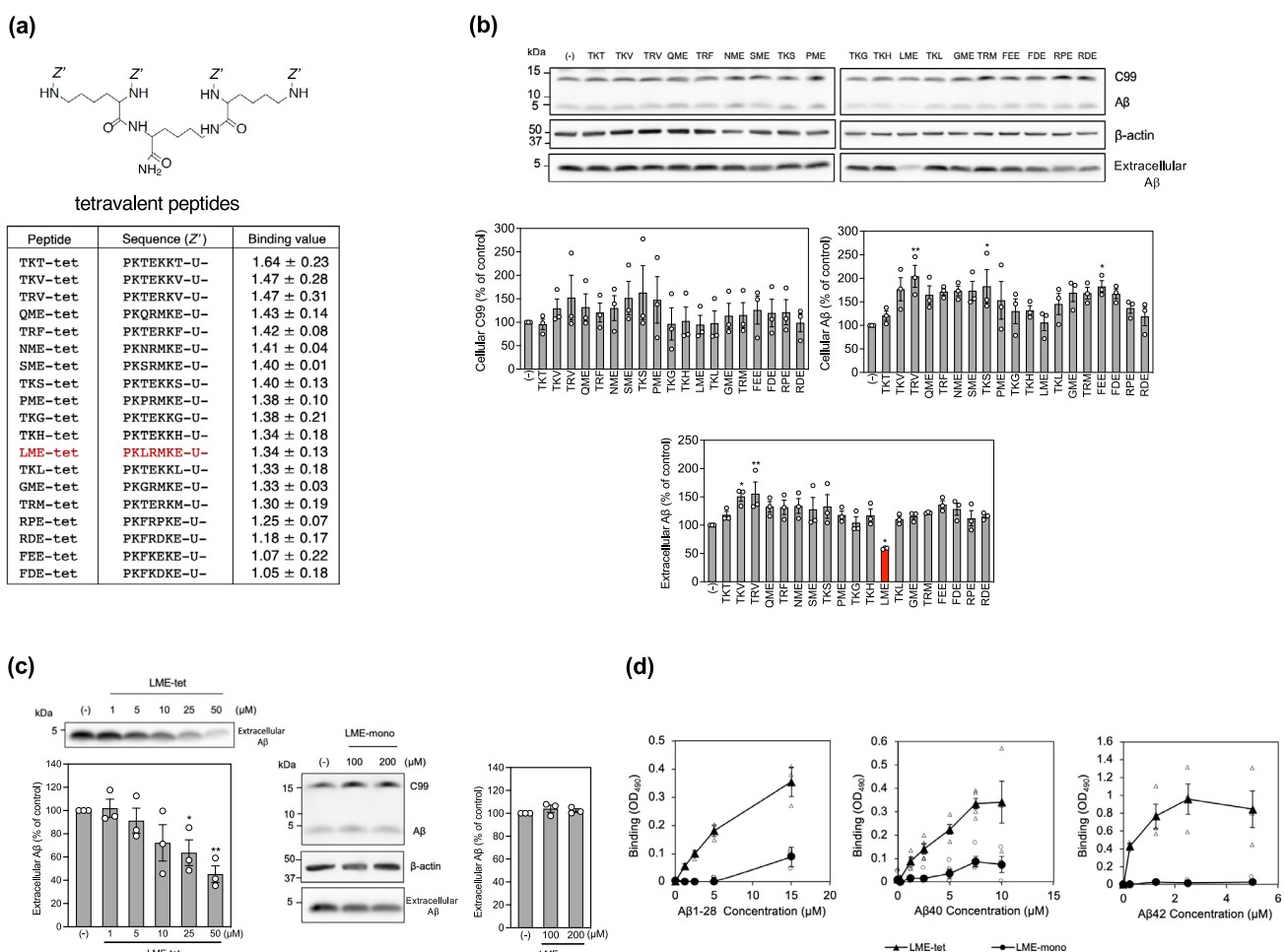

**Fig. 2 LME-tet efficiently inhibits production of Aβ in cells through a multivalent interaction. a** Schematic showing the structure of the tetravalent peptides and the Aβ1–28-binding motifs identified by the seventh round of screening. **b** Effects of tetravalent peptides on production of Aβ. 7WD10 Chinse Hamster Ovary (CHO) cells were treated with each tetravalent peptide (50 μM) for 48 h; cell lysates (for intracellular C99 and Aβ) and culture medium (for extracellular Aβ) from peptide-treated and vehicle control-treated cells were analyzed by western blot. Data are graphed as the percentage of the control value, showing the mean ± SE; n = 3. Significance vs. vehicle was calculated by analysis of variance (ANOVA), followed by one-sided Dunnett's test; *P < 0.05, **P < 0.01. **c** Effect of increasing concentrations of LME-tet on Aβ production (left panel). 7WD10 cells were treated with vehicle control or LME-tet at the indicated concentrations for 48 h, and the culture medium was analyzed by western blot. Data are graphed as the percentage of the control value, showing the mean ± SE; n = 3. Significance vs. vehicle was calculated by ANOVA, followed by one-sided Dunnett's test; *P < 0.05, **P < 0.01. Effect of LME-mono on Aβ production (right panels). 7WD10 cells were treated with vehicle control or LME-mono at the indicated concentrations for 48 h; cell lysates (for intracellular C99 and Aβ) and culture medium (for extracellular Aβ) were analyzed by western blot. LME-mono concentrations of 100 and 200 μM contain the same moles of binding motif as LME-tet at 25 and 50 μM, respectively. Data are graphed as the percentage of the control value, showing the mean ± SE; n = 3. Significance vs. vehicle was calculated by ANOVA, followed by one-sided Dunnett's test; *P < 0.05. **d** Binding of LME-tet and LME-mono to Aβ1-28, Aβ40 and Aβ42 was measured by enzyme-linked immunosorbent assay (ELISA). Values represent the mean ± SE; n = 3 for Aβ1–28 and Aβ42, n = 4 for Aβ40.

(Fig. 3b, upper left panel). Consistent with this observation, immunofluorescent staining of APP present in the plasma membrane co-localizes well with that of LME-tet (Fig. 3b, right panel), suggesting that LME-tet forms a complex with APP in the plasma membrane. Further, after subsequent incubation at 37 °C, this complex is endocytosed (Fig. 3b, right panel) and co-localizes with early endosome antigen 1 (EEA1), a well-known early endosomal marker (Fig. 3b, lower left panel), with a similar staining pattern to that was observed in the absence of LME-tet. These results indicate that LME-tet initially binds to APP present in the plasma membrane and is then incorporated into endosomes in a complex with APP.

We then examined whether LME-tet stimulates lysosomal degradation and/or autophagy to inhibit the Aβ production, by staining for lysosome-associated membrane glycoprotein 1

(LAMP1) or with Lysotracker, which are makers for the lysosome and acidic compartment, respectively. Our results indicate that 1 h after endocytosis of APP, only faint co-localization between each marker and APP is observed, and colocalization is not further increased in response to complex formation between APP and LME-tet (Fig. 3c and Supplementary Fig. 3). We further find that in the absence of peptides, treatment with lysosomal protease inhibitor (PI) slightly increases the amounts of intracellular APP and C99 (Fig. 3d), indicating that a small portion of intracellular APP and C99 is degraded in the lysosome or by basal autophagy. Consistently, levels of the autophagosomal markers p62 and LC3-II are increased by PI-treatment (Fig. 3d). However, treatment with LME-tet or LME-mono does not affect the PI-dependent increase of intracellular APP, C99, Aβ, p62 and LC3-II. Furthermore, the decreased accumulation of extracellular Aβ

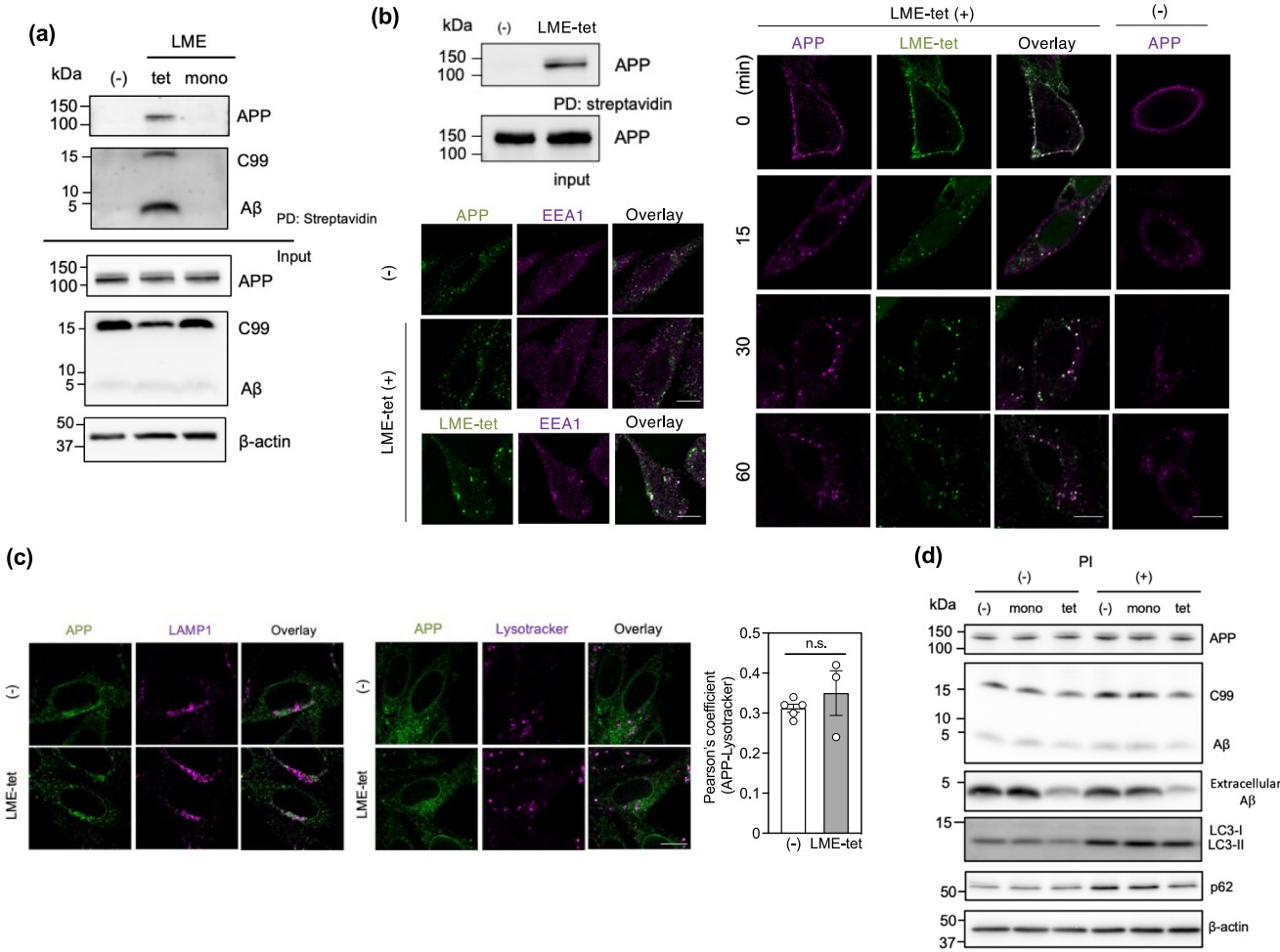

**Fig. 3 LME-tet forms a complex with APP and co-localizes with APP in endosomes. a** Binding of LME-tet to intracellular APP, C99, and Aβ. 7WD10 cells were treated with biotinylated LME-tet (50 µM) or LME-mono (200 µM) for 30 min at 37 °C. Cell lysates were then incubated with streptavidin beads, and coprecipitating proteins were analyzed by western blot; PD, pull down. **b** Binding of LME-tet to APP in the plasma membrane (upper left panel). 7WD10 cells were treated with biotinylated LME-tet (50 µM) on ice for 30 min, and cell lysates were analyzed as described above. Intracellular localization of APP-LME-tet complex (right panel). 7WD10 cells were treated with mouse monoclonal anti-human APP antibody 6E10 in the presence or absence of biotinylated LME-tet (50 µM) on ice for 30 min; cells were then washed and incubated at 37 °C for the indicated time. APP was detected by Alexa Fluor 546-conjugated goat anti-mouse IgG antibody, and LME-tet was detected by Alexa Fluor 488-conjugated streptavidin. Colocalization with early endosome antigen 1 (EEA1; lower left panel). 7WD10 cells were treated as described above, and after 15 min incubation at 37 °C, immunostaining for EEA1 was performed using rabbit polyclonal anti-EEA1 antibody, followed by Alexa Fluor 488-conjugated goat anti-rabbit IgG antibody. The scale bars indicate 10 µm. **c** APP co-localization with lysosome-associated membrane glycoprotein 1 (LAMP1), a lysosomal marker, was analyzed by immunocytochemical staining with rabbit polyclonal anti-LAMP1 antibody, followed by Alexa Fluor 488-conjugated goat anti-rabbit IgG antibody (left panel). Acidification of APP-containing endosomes was analyzed using Lysotracker (middle panel). In both cases, 7WD10 cells were treated with or without LME-tet (25 µM) for 60 min at 37 °C, and APP was detected as described above. The scale bars indicate 10 µm. Pearson's coefficient for colocalization between APP and Lysotracker is shown (right panel; mean ± SE, n (number of cells) = 10–24 or 7–14, for the absence (−) from five independent experiments or presence of LME-tet from three independent experiments, respectively). Pearson's coefficient was calculated using the Coloc 2 plugin in Image J. The significance *vs.* LME-tet (−) was calculated using ANOVA, followed by a one-sided Dunnett's test. n.s. not significant. **d** Effect of protease inhibitor (PI) on levels of APP, C99, Aβ, and the autophagosomal markers LC3 and p62, in LME-tet- and LME-mono-treated cells. 7WD10 cells were treated with LME-tet (25 µM) or LME-mono (100 µM) in the presence or absence of lysosomal PI cocktail for 48 h. Cell lysates (for intracellular APP, C99, Aβ, LC3, and p62) and culture medium (for extracellular Aβ) were analyzed by western blot.

observed in response to LME-tet is not affected by PI-treatment (Fig. 3d), further supporting the conclusion that neither lysosomal degradation nor autophagy is involved in the inhibition of Aβ production by LME-tet.

**LME-tet specifically blocks β-cleavage to inhibit production of Aβ.** Because the endosome is the primary site for APP processing, we next determined the effects of LME-tet on cleavage of APP by α-secretase and β-secretase. To this end, 7WD10 cells were treated with LME-tet for 48 h, and levels of intracellular APP, as

wells extracellular sAPPα and sAPPβ, which are produced by α- and β-secretase, respectively, were measured by western blot. We found that the total amounts of glycosylated APP (mature form) and the immature form were increased by 1.6 and 1.3 times, respectively, by LME-tet treatment (Fig. 4a). Further, although levels of extracellular sAPPα are not affected by LME-tet, extracellular sAPPβ is decreased by LME-tet in a dose-dependent manner (Fig. 4b, left and right panels, respectively). These results indicate that complex formation between APP and LME-tet in the plasma membrane does not affect cleavage by α-secretase, but the

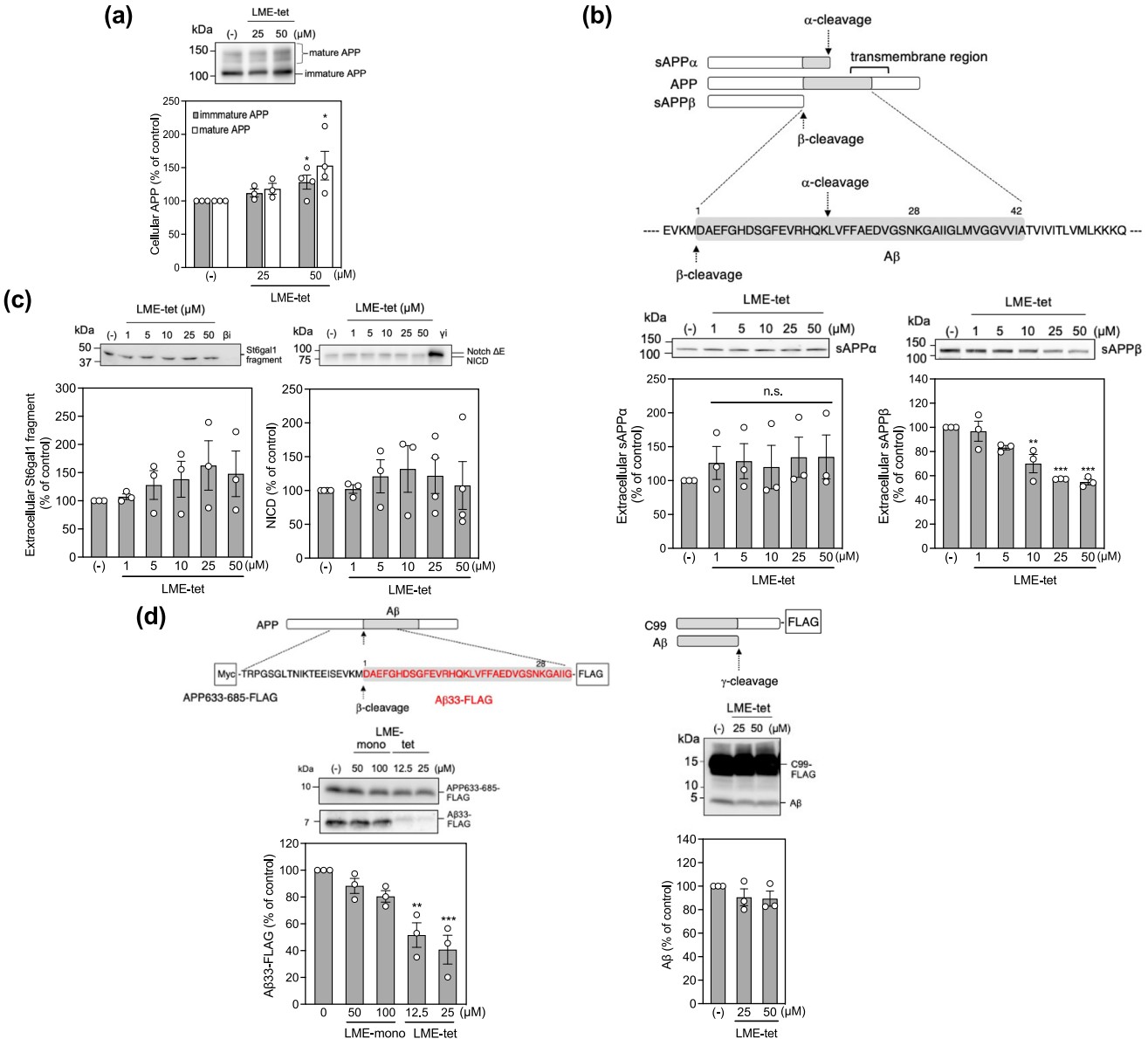

**Fig. 4 LME-tet specifically inhibits β-secretase to block production of Aβ. a** Effect of LME-tet on APP production. 7WD10 cells were treated with vehicle control or LME-tet at the indicated concentrations for 48 h. Cell lysates were analyzed by western blot. **b** Effect of LME-tet on production of extracellular sAPPα and sAPPβ Schematic showing α- and β-cleavage sites (top). 7WD10 cells were treated as in (**a**) and sAPPα and sAPPβ in culture medium were analyzed by western blot (bottom left and right panels, respectively). **a, b** APP and sAPPα were detected using anti-APP antibody 6E10, and sAPPβ was detected using anti-sAPPβ antibody. Data are graphed as the percentage of the control value, showing the mean ± SE; $n = 3$–4. Significance vs. vehicle was calculated by ANOVA, followed by one-sided Dunnett's test; $*P < 0.05$, $**P < 0.01$, $***P < 0.001$. **c** Effect of LME-tet on β-cleavage of β-galactoside α-2,6-sialyltransferase 1 (St6gal1) and γ-cleavage of NotchΔE. 7WD10-NotchΔE-St6gal1 cells were treated with LME-tet at the indicated concentrations and β-secretase inhibitor II (βi; 10 μM) for 48 h; culture medium was analyzed by western blot using anti-St6gal1 antibody (left panel). 7WD10-NotchΔE cells were treated with LME-tet at the indicated concentrations and γ-secretase inhibitor DAPT (γi; 10 μM) for 48 h; cell lysates were analyzed by western blot using anti-Myc antibody (right panel). Data are graphed as the percentage of the control value, showing the mean ± SE; $n = 3$–4. Significance vs. vehicle was calculated by ANOVA, followed by one-sided Dunnett's test. **d** Effect of LME-tet on β- and γ-secretase activities in vitro. Schematics showing β- and γ-cleavage sites/products and FLAG-tag locations (top left and right panels, respectively). Purified FLAG-tagged APP fragment (APP633–685-FLAG) with N-terminal Myc and C-terminal FLAG tags was incubated with β-secretase and LME-tet/LME-mono at the indicated concentrations for 4 h (left). FLAG-tagged C99 was incubated with γ-secretase and LME-tet at the indicated concentrations (right). Aβ33-FLAG (left) and Aβ (right) were measured by western blot using anti-Aβ antibody 82E1. APP633–685-FLAG (left) was measured by western blot using anti-Aβ antibody 6E10. Graphs show the mean ± SE, $n = 3$. Significance vs. vehicle was calculated by ANOVA, followed by one-sided Dunnett's test; $**P < 0.01$, $***P < 0.001$.

endocytosed complex within the endosome is resistant to cleavage by β-secretase.

We then evaluated whether the inhibitory effect of LME-tet is specific to cleavage of APP by β-secretase, by measuring the effects of LME-tet treatment on processing of other intracellular substrates for β- and γ-secretase. β-galactoside α-2,6-sialyltransferase 1 (St6gal1), a representative intracellular β-secretase substrate, is known to be cleaved by β-secretase to produce a St6gal1 fragment, which is released from cells[30]. We therefore treated 7WD10 cells stably expressing St6gal1 (7WD10-Notch

ΔE-St6gal1 cells) with LME-tet for 48 h and measured the amounts of extracellular St6gal1 fragment. Results show that production of extracellular St6gal1 fragment is not affected by LME-tet treatment but is completely suppressed by a β-secretase inhibitor II, which directly inhibits the catalytic activity of this enzyme (Fig. 4c, left panel). Similarly, we tested whether LME-tet affects cleavage of the representative intracellular γ-secretase substrate Notch, which is cleaved by γ-secretase to produce Notch intracellular domain (NICD)[31]. We found that treatment of 7WD10 cells stably expressing NotchΔE (7WD10-NotchΔE cells) with LME-tet for 48 h does not affect the production of NICD, whereas the γ-secretase inhibitor DAPT efficiently inhibits NICD production (Fig. 4c, right panel). These results show that the inhibitory effect of LME-tet is specific to cleavage of APP by β-secretase.

To further confirm the specificity of the inhibitory effect of LME-tet on APP, we measured the direct effects of LME-tet on β- and γ-secretase activities in vitro. For assessing β-secretase activity, we measured production of β-cleaved C-terminal fragment (Aβ33-FLAG) from a FLAG-tagged fragment of human APP (residues 633–685 in APP751), which contains the Aβ1−28 region. We found that LME-tet, but not LME-mono, efficiently inhibits production of Aβ33-FLAG (Fig. 4d, left panel), indicating that direct binding of LME-tet to APP633-685-FLAG via a multivalent interaction blocks β-secretase from accessing APP633–685-FLAG. In contrast, LME-tet did not show any inhibitory effects on the production of Aβ from FLAG-tagged C99 (Fig. 4d, right panel), indicating that LME-tet specifically inhibits β-cleavage.

**LME-tet inhibits the conformational change and aggregation of Aβ42**. Based on our observation that LME-tet efficiently binds to both synthetic Aβ and intracellular Aβ, we next determined whether LME-tet can also bind to extracellular Aβ released from cells. To address this question, conditioned medium from 7WD10 cells was subjected to coprecipitation assays using biotinylated LME-tet. We found that total Aβ, including Aβ42, was coprecipitated with LME-tet, but not with LME-mono, indicating that LME-tet potently binds to Aβ released from cells via multivalent interaction (Fig. 5a).

These findings, combined with our results showing that LME-tet displays preferential binding affinity for the aggregation-prone Aβ42 among all Aβ species tested, next prompted us to examine whether LME-tet can interfere with the conformational change and aggregation of Aβ42. Using circular dichroism (CD) analysis, we found that LME-tet markedly delays the conformational change of Aβ42, characterized by the formation of β-sheet structures, which is a hallmark of Aβ42 oligomerization (Fig. 5b). In contrast, under the same conditions, no effect was observed with LME-mono. This inhibitory effect on conformational change of Aβ42 causing oligomerization is further evident from molar ellipticity analysis showing a decreased velocity of conformational change for Aβ42 incubated with LME-tet (Fig. 5b, lower right panel). Consistent with these findings, we observe a clear inhibitory effect of LME-tet, but not LME-mono, on fibril formation of Aβ42, as measured by the Thioflavin T-binding assay (Fig. 5c) and electron microscopy (Fig. 5d). Furthermore, the effect of LME-tet on fibril formation was examined at various time points by electron microscopy (Supplementary Fig. 4). In the absence of LME-tet, fibril formation was detected after 4 h of incubation; additionally, tangled fibrils and subsequent aggregation of Aβ42 were observed after 8 h of incubation. LME-tet almost completely inhibited fibril formation for up to 24 h of incubation, which is consistent with the Thioflavin T-binding assay results. Collectively, these data show that LME-tet can

suppress the conformational change, fibril formation, and subsequent aggregation of Aβ42 through multivalent interaction with this peptide. These inhibitory effects were not observed or were substantially weaker with QME-tet, NME-tet, SME-tet, GME-tet, or PME-tet, indicating the importance of Leu at position 3 for efficient inhibition (Supplementary Fig. 5). Further, using the Thioflavin T-binding assay, we found that LME-tet can potently impede the enhanced growth of Aβ42 fibril that are observed in the presence of a small amount of Aβ42 fibril-seed (Fig. 5e), an inhibitory effect that is not observed with the other peptides (Supplementary Fig. 6). Thus, combined with the results of Fig. 5c and Supplementary Fig. 4, LME-tet can inhibit the initial formation of Aβ42 fibril-seed, which markedly stimulates the subsequent growth of the fibrils.

Previously, prefibrillar aggregates or oligomers of Aβ42, but not fully aggregated fibrils, were shown to have cytotoxic activity against neuronal cells[32,33]. We examined the protective effect of LME-tet on the cytotoxic activity of Aβ42 prefibrillar aggregates using the Neuro-2a neuronal cell line. Incubation of the cells with Aβ42 for 48 h in the absence of LME-tet reduced cell viability by 20%. The presence of LME-tet substantially rescued the viability by 5% (Supplementary Fig. 7), indicating that LME-tet can potentially protect cells by retarding the formation of cytotoxic prefibrillar aggregates or oligomers, consistent with the above observations (Fig. 5 and Supplementary Fig. 4).

**LME-tet reduces Aβ levels in the brain of AD model mice**. Lastly, to determine the effect of LME-tet treatment on levels of Aβ in the brain, we used AD model mice with a single humanized *App* knock-in (KI), which carries the Swedish (NL), Beyreuther/Iberian (F), and Arctic (G) mutations (*App*[NL-G-F/NL-G-F] mice). These mice show accumulation of Aβ in the brain and display pathological features of AD[32]. For these experiments, to increase stability of LME-tet in mice, we used a protease-resistant N-terminal acetylated form of LME-tet (Ac-LME-tet), which shows a similar inhibitory effect on Aβ production in 7WD10 cells (Supplementary Fig. 8). Mice were then intraperitoneally treated with phosphate-buffered saline (PBS) or Ac-LME-tet in PBS (71 mg/kg), and after 7 days, their brains were analyzed. Notably, we detected substantial reductions in the Aβ levels in the cortex and the olfactory bulb of mice treated with Ac-LME-tet compared with that in control-treated animals, indicating that Ac-LME-tet can function in vivo (Fig. 6). To investigate whether Ac-LME-tet penetrates the brain, we intraperitoneally administered FITC-labeled LME-tet and examined the brain distribution of the peptide. Although no peptide signals were observed in individual neurons, we detected substantial peptide signals in the choroid plexus of the lateral ventricle (CPLV) of the brain (Supplementary Fig. 9). Thus, a certain amount of the peptide may penetrate the blood-brain barrier of the CPLV and enter the brain. In addition, we compared the efficiency of LME-tet incorporation into cells with that of another identified peptide, PME-tet, which does not inhibit Aβ production (Fig. 2b). We found that LME-tet can be incorporated with much higher efficacy than PME-tet (Supplementary Fig. 10). This feature of LME-tet to be easily transported to cells through the cellular membrane, may contribute to its efficient localization in the brain.

On the other hand, the amount of Aβ in the plasma was not affected (Supplementary Fig. 11). Recently, peripheral Aβ was shown to be produced by adipose tissues, skeletal muscles, and the liver and function as a negative modulator of insulin secretion[34]. Thus, our results indicate that Ac-LME-tet does not affect the physiological function of peripheral Aβ as a regulator of insulin secretion, further emphasizing the advantage

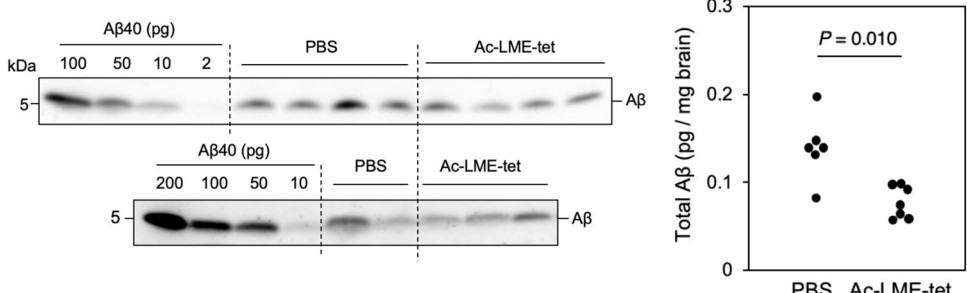

**Fig. 5 LME-tet inhibits the conformational change and subsequent fibrillation of Aβ42. a** Binding of LME-tet to extracellular Aβ. Culture medium from 7WD10 cells recovered after 48 h of growth was treated with biotinylated LME-tet (50 μM) or LME-mono (200 μM). Medium was then incubated with streptavidin beads, and coprecipitating Aβ40 and Aβ42 were analyzed by western blot using anti-Aβ or anti-Aβ42-specific antibody. **b** Time-dependent conformational change in Aβ42 (20 μM) in the presence of LME-tet (2 μM, lower left panel) or LME-mono (2 μM, upper right panel). Far-UV circular dichroism (CD) spectra (wavelength 190–250 nm) were measured at the indicated time points. Kinetic rates of β-sheet formation of Aβ42 were estimated by measuring time-dependent alterations of molar ellipticities of CD spectra at 217 nm (lower right panel). Data represent the mean ± SE; $n = 3$-4. **c** Fibrillation profiles of Aβ42 in the presence of LME-tet (2 μM) or LME-mono (2 μM), measured by Thioflavin T-binding assay. Time-dependent increments of Aβ42–Thioflavin T binding are shown. Data represent the mean ± SE; $n = 3$. **d** Fibrillation profiles of Aβ42 in the presence of LME-tet (2 μM) or LME-mono (2 μM), analyzed by electron microscopy; representative images at ×10 K, ×50 K, and ×200 K magnification (left, right, and center, respectively) are shown. Scale bars indicate 2 μm, 400 nm, and 100 nm, respectively. **e** Effects of LME-tet (2 μM) on growth of Aβ42 fibril in the presence of 2% (v/v) Aβ42 fibril-seed, measured by the Thioflavin T-binding assay. Time-dependent increments of Aβ42–Thioflavin T binding are shown. Data represent the mean ± SE; $n = 3$.

**Fig. 6 Ac-LME-tet reduces Aβ levels in the brain of AD model mice.** $App^{NL\text{-}G\text{-}F/NL\text{-}G\text{-}F}$ mice were intraperitoneally administered N-terminal-acetylated LME-tet (Ac-LME-tet) or phosphate-buffered saline (PBS) control. After 7 days, mice were then sacrificed, and the amounts of human Aβ present in the cerebral cortex and the olfactory bulb were measured by western blot using anti-human Aβ antibody 4G8. The indicated amounts of Aβ were used for quantification; $n = 6$ for control, and $n = 7$ for Ac-LME-tet-treated mice. Significance was determined by the Mann–Whitney $U$ test; $P = 0.010$ for Ac-LME-tet-treated *vs.* PBS-treated mice.

of Ac-LME-tet for use as an effective therapeutic compound for AD treatment.

## Discussion

In this study, we developed a new technique for affinity-driven screening of tetravalent peptide libraries synthesized on a cellulose membrane and used this to identify a tetravalent peptide, LME-tet, which binds to APP and inhibits Aβ production in cells based on the clustering effect. Our lab previously pioneered the use of a multivalent peptide library screening technique, in which high-affinity binding motifs that function based on the clustering effect are identified by performing amino acid sequencing of randomized peptides specifically bound to the target molecule[22,23,25–27]. Compared to this technique, our new method is unique in that at every screening stage, the optimal amino acid at each position can be determined with high resolution. This strategy allowed the determination of a wide variety of candidate peptide with high-affinity binding activities to Aβ1−28. One of these peptides, LME-tet, was found to be a highly efficient inhibitory compound relative to the other candidate compounds, including QME-tet, NME-tet, SME-tet, GME-tet, and PME-tet, all of which have only one amino acid difference at position 3 compared to LME-tet. These findings clearly demonstrate the advantage of this new method for critically optimizing the functional binding motif of tetravalent peptides. Furthermore, the functional motif of LME-tet could not have been obtained by other known techniques, such as monomeric random-peptide library screening and phage display library screening, as these can only be used to identify monomeric peptides, which theoretically cannot exert the clustering effect.

Our findings here are consistent with a model in which LME-tet binds to Aβ1−28, Aβ40, and Aβ42 based on the clustering effect, as the monovalent peptide with the same binding motif, LME-mono, is unable to bind these molecules. Although the precise mechanism for LME-tet binding remains to be elucidated, it may firstly bind to one target molecule through a single binding motif, and then sequentially induce cluster formation through multivalent interaction to enhance the apparent binding affinity, sometimes referred to as avidity. In previous studies, this type of high-avidity intermolecular interaction has been observed between multivalent ligands, including multivalent peptides or carbohydrate-polymers, and target molecules prone to cluster or that function via multi-subunit structures[22,27,29,35,36]. Thus, we expect that the higher binding affinity of LME-tet to Aβ42 than to Aβ40 may reflect the propensity of the former to readily form oligomers, such as dimers, trimers, and multimers, in which each LME-tet binding site is clustered in close proximity to exert a clustering effect on the binding.

In cells, we further found that LME-tet initially binds to APP present in the plasma membrane and is incorporated into endosomes in a complex with APP. Consistent with our previous report[37], a small fraction of this incorporated APP is then degraded in the acidic compartment (i.e., in lysosomes and/or autophagosomes), although LME-tet does not enhance this process to inhibit Aβ production (Fig. 3d). Rather, LME-tet specifically blocks the β-cleavage of APP, but not the γ-cleavage, thereby inhibiting production of Aβ (Fig. 7). Recently, it was shown that dimerization of APP, which is induced by cholesterol-lowering drugs such as statins, inhibits interaction between APP and β-secretase in the membrane and consequently reduces Aβ production[38]. This observation is consistent with another previous report showing that APP can directly bind to cholesterol through its transmembrane domain, which contains three GXXXG motifs, to enhance interaction between APP and β-secretase[39]. We speculate that LME-tet, but not LME-mono, may use a similar mechanism as that employed by statins to cluster APP through multivalent interactions in the membrane and disrupt interaction with β-secretase. This further supports the importance of the state of APP molecules present in the membrane for regulating β-secretase and ultimately, modulating Aβ production.

A variety of β-secretase inhibitors that directly bind to this enzyme and inhibit its catalytic activity have been developed. However, in clinical trials, these compounds were found to display severe off-target effects, such as worsening cognitive function and hepatic toxicity[10,11]. These are inevitable due to the fact that β-secretase cleaves a range of physiologically important substrates other than APP, including neuregulin (NRG)[40–42] and β-galactoside α2, 6-sialyltransferase[43,44]. Notably, sterol-linked β-secretase inhibitors, which target endosomal β-secretase, have been shown to block cleavage of APP, but not non-amyloid substrates, such as NRG or the axon guidance molecule L1[45,46]. Use of such hybrid inhibitors, however, cannot exclude the possibility that they may inhibit β-cleavage of other non-amyloid substrates, as they still target the β-secretase enzyme. In contrast, by targeting the ectodomain of C99, rather than β-secretase or γ-secretase itself, we previously identified a set of monomer peptides with the CDCYCXXXXCXCXSC motif that inhibit β- and γ-cleavage by binding to APP and C99, respectively, and thus inhibit Aβ production in cells[47]. Interestingly, here we found that LME-tet, which also targets the ectodomain, inhibits Aβ production by specifically blocking the β-cleavage of APP, but not the γ-cleavage of C99. Furthermore, in stark contrast to LME-tet, the best inhibitor among the previously identified ectodomain-targeting peptides (peptide #4) did not inhibit the conformational change of Aβ42 through β-sheet formation (Supplementary Fig. 12). This indicates that LME-tet binds to APP in a manner that is quite distinct from peptide #4. Thus, LME-tet represents a novel type of APP-targeting inhibitor that specifically inhibits β-cleavage of APP, as well as the downstream oligomerization of Aβ.

As noted above, LME-tet binds more tightly to Aβ42 (apparent Kd = 0.20 μM) than to Aβ40 (apparent Kd = 2.7 μM) in vitro and also efficiently binds Aβ42 released from cells. Of these two species, Aβ42 shows a greater propensity for aggregation, which leads to formation of the amyloid fibrils observed in senile plaques[48]. The recently published fibril structure of Aβ42 as determined by cryo-electron microscopy shows that residues 1–9, 11–21, and 23–33 adopt β-sheet conformations to form filaments comprising Aβ42 molecules stacked in a parallel cross-β structure[17]. Here, we found that LME-tet delays this conformational change in Aβ42 characterized by β-sheet formation, which in turn, inhibits Aβ42 aggregation. This suggests a model wherein multivalent binding of LME-tet to Aβ molecules leads to formation of a complex that cannot oligomerize to form fibrils due to disruption of the β-sheet conformation (Fig. 7). In addition, electron microscopic analysis showed that LME-tet clearly inhibited fibril formation even after 4 h of incubation, further supporting the contention that LME-tet may function at an early stage of pre-fibril formation (Supplementary Fig. 4). Previous studies showed that a series of monomeric peptides containing a cationic Arg-cluster or amyloidogenic sequences derived from Aβ, such as the KLVFF (Aβ16–20), GGVVIA (Aβ37–42) and LPFFD (modified Aβ17–21) motifs, act as aggregation inhibitors against Aβ42[49–52]. The recently developed and most effective peptide with a modified KLVFF motif, Ac-KL-$_N$VF-$_N$FA-NH$_2$, which is a diaminated derivative, inhibited fibrillation of Aβ42 by 40 % with an equivalent molar concentration after 8 h of incubation as measured by the Thioflavin T-binding assay. Additionally, LME-tet completely inhibited fibrillation, even with 1/10 of the molar concentration of Aβ42, after 8 h of incubation (Fig. 5c)[53]. Some of the peptides with Arg-cluster, which function as effective as aggregation inhibitors in vivo, have 5 to 8

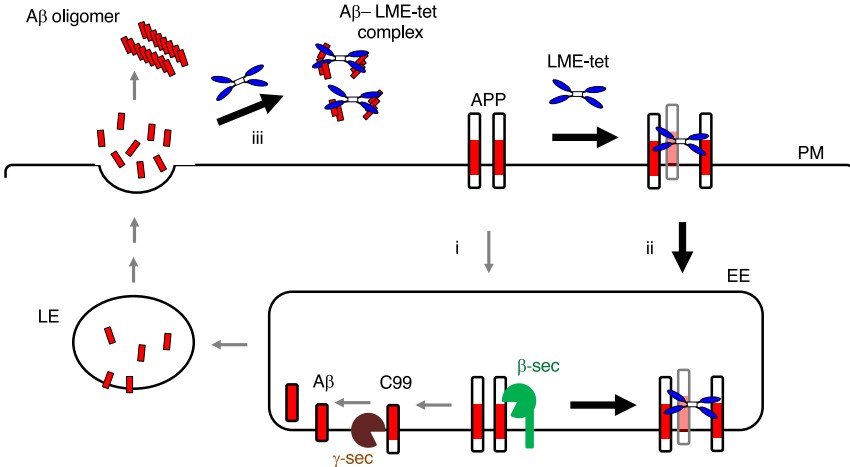

**Fig. 7 Model for the proposed molecular mechanism underlying the intracellular and extracellular effects of LME-tet.** APP present in the plasma membrane (PM) is internalized into early endosomes (EE) and then into late/recycling endosomes (LE), where Aβ is produced by sequential proteolysis of APP performed by β-secretase and γ-secretase. Aβ is released into the extracellular space and then aggregates to from Aβ oligomers (i, gray arrows). LME-tet binds to APP in the plasma membrane through multivalent interaction. This complex is transported to the endosomes and blocks Aβ production through specific inhibition of β-cleavage, but not γ-cleavage (ii). In addition, LME-tet suppresses oligomerization of extracellular Aβ by inhibiting the formation of its β-sheet conformation (iii).

consecutive Arg residues, consistent with the finding that increasing cationic charge is associated with greater Aβ-binding affinity[54]. In contrast, LME-tet has a tetravalent structure with one Arg in the binding motif, making it a unique peptide-based aggregation inhibitor.

Lastly, we tested the function of LME-tet in vivo and show that a stabilized form of this peptide can efficiently reduce Aβ levels in the brain of $App^{NL-G-F/NL-G-F}$ mice, suggesting that LME-tet may be efficacious for the treatment of AD. This is further supported by our finding that LME-tet binding to APP inhibits cleavage by β-secretase—the rate-limiting protease in Aβ production and a particularly attractive drug target[15]. Notably, a recent study reported that pharmacological inhibition of β-secretase is effective for limiting the initial process of plaque formation in vivo, but not for preventing subsequent plaque growth[55]. Thus, a key added benefit of LME-tet lies in its ability to inhibit, not only the formation of Aβ, but also its aggregation and polymerization, with high efficacy (Fig. 7). Thus, LME-tet may hold potential as a unique dual-function therapeutic compound for treatment of AD.

## Methods

**Antibodies**. The following antibodies were obtained and used at the indicated dilution as follows: mouse anti-human APP antibody 6E10 (1:250, Biolegend, San Diego, CA, USA, Cat# 803001), mouse anti-human Aβ antibody 82E1 (1:500, Immuno-Biological Laboratories Co., Ltd., Gunma, Japan, Cat# 10323), rabbit anti-human Aβ42 specific antibody (1:1000, Immuno-Biological Laboratories Co., Ltd., Cat#18582), mouse anti-Aβ17-24 antibody 4G8 (1:1000, Biolegend, Cat#800709), rabbit anti-sAPPβ antibody (1:50, Immuno-Biological Laboratories Co., Ltd., Cat# 18957), rabbit anti-EEA1 antibody (1:500, Thermo Fisher Scientific, Waltham, MA, USA, Cat# PA1-063A), rabbit polyclonal anti-LAMP1 antibody (1:200, Abcam, Cambridge, UK, Cat#ab24170), rabbit polyclonal anti-SQSTM1/p62 antibody (1:1000, MBL International, Woburn, MA, USA, Cat#PM045), rabbit polyclonal anti-LC3 antibody (1:1000, MBL International, Cat#PM036), rabbit polyclonal anti-β-actin antibody (1:2000, MBL International, Cat# PM-053), rabbit anti-Sialyltransferase 1 (St6gal1) antibody (1:100, Immuno-Biological Laboratories Co., Ltd., Cat# 18983), mouse anti-Myc-Tag antibody 9B11 (1:1000, Cell Signaling Technology, Danvers, MA, USA, Cat# 2276), horseradish peroxidase (HRP)-conjugated goat anti-rabbit IgG antibody (1:5000, Cell Signaling Technology, Cat# 7074 S), HRP-conjugated horse anti-mouse IgG antibody (1:2000, Cell Signaling Technology, Cat# 7076 S), Alexa Fluor 488-conjugated goat anti-mouse IgG antibody, (1:2000, Thermo Fisher Scientific, Cat# A-11001), Alexa Fluor 546-conjugated goat anti-rabbit IgG antibody, (1:2000, Thermo Fisher Scientific, Cat# A-11010), Alexa Fluor 546-conjugated goat anti-mouse IgG antibody (1:2000, Thermo Fisher Scientific, Cat# A-11003).

**Synthesis and screening of tetravalent peptide libraries**. Tetravalent peptide libraries were synthesized on a cellulose membrane, using a ResPep SL SPOT synthesizer (Intavis Bioanalytical Instruments AG, Kölln-reisiek, Schleswig-Holstein, Germany)[56]. Fmoc-βAla-OH (Watanabe Chemical Industries, Hiroshima, Japan) was used in the first cycle, followed by aminohexanoic acid as a spacer. Fmoc-Lys(Fmoc)-OH (Watanabe Chemical Industries) was used in the next two cycles to create four branches in the peptide chain for subsequent motif synthesis. Membranes were blocked with 5% skim milk and blotted with biotinylated Aβ1–28 (10 μg/ml) (AnaSpec, Fremont, CA, USA). After extensive washing, the Aβ1–28 bound to each tetravalent peptide was detected with horseradish peroxidase (HRP)-labeled streptavidin (Agilent Technologies, Santa Clara, CA, USA) and Enhanced Chemiluminescence (ECL) Plus Reagent (PerkinElmer, Inc, Waltham, MA, USA). The intensity of each peptide spot was then quantified as a pixel value, using an ImageQuant LAS 500 CCD imager (GE Healthcare Life Sciences, Marlborough, MA, USA).

**Peptides**. Tetravalent and monomeric peptides were synthesized using N-α-Fmoc-protected amino acids and standard BOP/HOB coupling chemistry[56]. The terminal amino groups of tetravalent peptides were biotinylated with biotin (Sigma-Aldrich, St. Louis, MO, USA) and 1-(bis[dimethylamino]methylene)-1H-benzotriazolium 3-oxide hexafluorophosphate (Peptide Institute Inc., Osaka, Japan) in the last cycle of peptide synthesis. The synthesized peptides were validated by mass spectrometry analysis, using the AutoflexII time-of-flight (TOF)/TOF system (Bruker Corp., Billerica, MA, USA).

**Cell culture**. Chinese Hamster Ovary (CHO) cells stably expressing human APP (APP1-751), referred to as 7WD10 cells, were maintained at 37 °C in Dulbecco's Modified Eagle Medium (DMEM; Sigma), supplemented with 10% fetal bovine serum (FBS), 50 units/ml penicillin, 50 μg/ml streptomycin, and 200 μg/ml G418 (Nacalai Tesque, Kyoto, Japan). 7WD10 cells stably expressing Myc-tagged Notch with a deletion in the extracellular region (NotchΔE)[57] (7WD10-NotchΔE cells) were maintained in DMEM, supplemented with 10% FBS, 50 units/ml penicillin, 50 μg/ml streptomycin, 200 μg/ml G418 (Nacalai Tesque), and 250 μg/ml Hygromycin B (Wako Chemicals, Richmond, VA, USA). 7WD10-NotchΔE cells stably expressing β-galactoside α-2,6-sialyltransferase 1 (St6gal1; 7WD10-Notch ΔE-St6gal1 cells) were maintained in DMEM, supplemented with 10% FBS, 50 units/ml penicillin, 50 μg/ml streptomycin, 200 μg/ml G418, 250 μg/ml Hygromycin B (Wako Chemicals), and 100 μg/ml Zeocin (InvivoGen, San Diego, CA, USA). All cell lines in this study were negative for mycoplasma contamination.

**Western blot**. 7WD10, 7WD10-NotchΔE, or 7WD10-Notch ΔE-St6gal1 cells were seeded in 24-well plates and cultured at 37 °C, in the presence or absence of the indicated concentrations of tetravalent or monomeric peptides. After 48 h incubation, culture medium and cells were recovered separately. Cells were lysed in lysis buffer, containing 2% sodium dodecyl sulfate (SDS), 80 mM Tris-HCl (pH 6.8), 7% glycerol, 250 mM dithiothreitol (DTT), and 0.025% phenol red. Proteins in culture medium and lysates were separated by tricine SDS-polyacrylamide gel (12% acrylamide) electrophoresis (SDS-PAGE) and transferred to nitrocellulose membranes (Wako Chemicals). To separate immature APP and mature APP, glycine

SDS-PAGE (7% acrylamide) was used. After boiling in phosphate-buffered saline (PBS) for 5 min and then blocking with 5% skim milk, the membranes were immunoblotted with the indicated primary antibodies, followed by HRP-conjugated secondary antibodies. Protein bands were visualized using ECL Plus Reagent (PerkinElmer) and analyzed using an ImageQuant LAS 500 CCD imager (GE Healthcare Life Sciences). Total amounts of target proteins were measured using Image J software (National Institutes of Health; NIH), after normalization to β-actin as an internal control.

**Enzyme-linked immunosorbent assay (ELISA) to measure binding between Aβ and LME peptides**. The indicated amounts of LME-tet (8.3 μM) or LME-mono (33 μM) were applied onto each well of a regular 96-well ELISA plate (Thermo Fisher Scientific) and incubated overnight for 24 h. After blocking, the plate was incubated with various concentrations of Aβ1–28, Aβ40, or Aβ42 (Peptide Institute) for 2 h at room temperature. Bound Aβ was detected using mouse monoclonal anti-human Aβ antibody 82E1 (Immuno-Biological Laboratories Co., Ltd.) and HRP-conjugated horse anti-mouse IgG antibody (Cell Signaling Technology).

**Coprecipitation assay using biotinylated peptides**. For coprecipitation from culture medium, 7WD10 cells in 12-well plates were cultured for 48 h at 37 °C, and the culture medium was recovered. This culture medium was treated with biotinylated LME-tet (50 μM) or LME-mono (200 μM) for 24 h at 4 °C, and then treated with streptavidin beads (Thermo Fisher Scientific) for 24 h at 4 °C. After extensive washing, the beads were analyzed by western blot, as described above. For coprecipitation from cell lysates, 7WD10 cells in 12-well plates were cultured for 30 min at 4 °C or 37 °C in the presence or absence of biotinylated LME-tet (50 μM) or LME-mono (200 μM). The cells were recovered and lysed in lysis buffer, as described above. Lysates were treated with streptavidin beads (Thermo Fisher Scientific) for 24 h at 4 °C, and after extensive washing, the beads were analyzed by western blot, as described above.

**Intracellular localization of APP and LME-tet**. Sub-confluent 7WD10 cells in 35-mm glass-bottom dishes were treated with mouse monoclonal anti-human APP antibody 6E10 (1:250 dilution) and biotinylated LME-tet (50 μM) on ice for 30 min. After washing, cells were incubated at 37 °C for the indicated times, fixed with 4% paraformaldehyde (PFA), and permeabilized with 0.2% TritonX-100. APP was detected using Alexa Fluor 546-conjugated goat anti-mouse IgG antibody. LME-tet was detected using Alexa Fluor 488-conjugated streptavidin. Early endosome antigen 1 (EEA1) was detected using rabbit polyclonal anti-EEA1 antibody, followed by Alexa Fluor 488-conjugated goat anti-rabbit IgG antibody. Lysosome-associated membrane glycoprotein 1 (LAMP1) was detected using rabbit polyclonal anti-LAMP1 antibody, followed by Alexa Fluor 488-conjugated goat anti-rabbit IgG antibody. For detection of the intracellular acidic compartment, cells were treated with 75 nM Lysotracker DND-99 (Molecular Probes, Eugene, OR, USA) for 1 h, followed by PFA fixation and permeabilization with 500 μg/ml digitonin (Wako Chemicals). Fluorescent images were analyzed using LSM700 laser scanning confocal microscopy (Zeiss, Oberkochen, Germany) and Image J software (NIH).

**Protease inhibitor treatment**. 7WD10 cells were treated with LME-tet (50 μM) or LME-mono (200 μM) in the presence or absence of lysosomal protease inhibitor cocktail containing 30 μM E-64 and 20 μM leupeptin for 48 h at 37 °C. The culture medium and cells were recovered separately, and proteins in each were analyzed by western blot, as described above.

**β-Secretase and γ-secretase assays in cells**. 7WD10-NotchΔE-St6gal1 cells were treated with LME-tet or β-secretase inhibitor II (βi; 10 μM) (Calbiochem, MilliporeSigma, Burlington, MA, USA) for 48 h. The culture medium was then analyzed by western blot, using anti-St6gal1 antibody to the detect β-cleavage product, St6gal1 fragment. 7WD10-NotchΔE cells were treated with LME-tet or γ-secretase inhibitor DAPT (γi; 10 μM) (Calbiochem, MilliporeSigma) for 48 h. The cell lysates were analyzed by western blot, using anti-Myc-Tag antibody to detect the γ-cleavage product, Notch intracellular domain (NICD).

**β-Secretase and γ-secretase assays in vitro**. A fragment of human APP (position 633–685 in APP751), with N-terminal Myc and C-terminal FLAG tags (APP633–685-FLAG) was expressed in *Escherichia coli* BL21 cells and affinity-purified using anti-FLAG M2 beads (Sigma)[45]. Purified APP633–685-FLAG (50 nM) was incubated with β-secretase (Sigma) for 4 h, according to the manufacturer's instructions, in the presence or absence of the indicated concentrations of LME-tet or LME-mono. β-cleaved C-terminal fragments (Ab33-FLAG) from APP633–685-FLAG were visualized by western blot with anti-human Aβ antibody 82E1 and quantified by Image J software (NIH).

γ-Secretase was prepared from the microsomal fraction of HEK293 cells, as previously described[58]. FLAG-tagged C99 was prepared using the Bac-to-Bac Baculovirus Expression System (Invitrogen, Thermo Fisher Scientific) and used as a γ-secretase substrate[47]. FLAG-tagged C99 was incubated with enzyme in dilution buffer, containing 12.5 mM PIPES (pH 7.2), 62.5 mM sucrose, 0.25 mM ethylene

glycol tetraacetic acid (EGTA), 0.25 mM diisopropyl fluorophosphate (DIFP), 2.5 μg/ml antipain, 2.5 μg/ml leupeptin, 2.5 μg/ml tosyl-L-lysyl-chloromethane hydrochloride (TLCK), 1.25 mM phenanthroline, 0.25 mM thiorphan, and 0.1% phosphatidylcholine at 37 °C for 4 h in the presence or absence the indicated concentrations of LME-tet. Production of Aβ in the reaction mixtures was measured by western blot analysis with anti-human Aβ antibody 82E1.

**Circular dichroism (CD) spectroscopy**. Aβ42 (20 μM) was incubated in the presence or absence of each peptide in 50 mM sodium phosphate buffer (pH 7.4) for the indicated time period at 37 °C. Far-UV CD spectra of Aβ42 were measured using a J-1500 spectrometer (JASCO Inc., Tokyo, Japan). Kinetic rates of β-sheet formation of Aβ42 were estimated by measuring time-dependent alterations of molar ellipticities of CD spectra at 217 nm[59,60].

**Thioflavin T-binding assay**. Aβ42 (20 μM) was incubated in the presence or absence of each peptide in 50 mM sodium phosphate buffer (pH 7.4) for the indicated time period at 37 °C. Each Aβ solution (10 μL) was added to 800 μL of 20 μM Thioflavin T in 20 mM Tris-HCl (pH 8.0), and then subjected to excitation at 450 nm. Emission was monitored at 482 nm, using a FP-8300 fluorescence spectrometer (JASCO Inc.)[54,59]. Effects of LME-tet on growth of Aβ42 fibrils was measured as follows. To prepare the Aβ42 fibril-seed, Aβ42 (20 μM) was incubated in 50 mM sodium phosphate buffer (pH 7.4) for 2 days at 37 °C. Aβ42 (20 μM) was then incubated in the presence or absence of LME-tet (2 μM) in 50 mM sodium phosphate buffer (pH 7.4), with or without 2% (v/v) of the fibril-seed for the indicated time period at 37 °C. Each Aβ solution was analyzed as described above.

**Transmission electron microscopy (TEM) analysis**. Aβ42 (20 μM) was incubated in the presence or absence of each peptide in 50 mM sodium phosphate buffer (pH 7.4) at 37 °C for 3 days. Fibril samples (10 μL) were adsorbed on 400-mesh collodion-coated copper grids for 10 min and negatively stained with 2% phosphotungstic acid. Morphology of fibrils was analyzed using a transmission electron microscope (TEM), H-7650 (Hitachi, Tokyo, Japan), with at an accelerating voltage of 80 kV.

**Treatment of mice with LME-tet**. Mice with single humanized *App* KI, carrying Swedish (NL), Beyreuther/Iberian (F), and Arctic (G) mutations (*App*NL-G-F/NL-G-F mice), were generated as described previously[32]. Mice were maintained under standard housing conditions, with lights on at 8:00 and lights off at 20:00, and animals were provided with water and food *ad libitum*. To increase stability of LME-tet in mice, acetylation of its terminal amino groups was performed using N-acetylimidazole, to generate Ac-LME-tet. Stabilized Ac-LME-tet in PBS (71 mg/kg) or PBS-only control was administered to 7-week-old mice intraperitoneally. After 7 days, mice were euthanized, and brains were removed. The cerebral cortex and olfactory bulb from one hemisphere were homogenized, extracted with a 4× volume of Tris-buffered saline solution (TBSs), containing 1% NP-40 and centrifuged at 100,000 × *g* for 1 h. The supernatant (1 ml) was subjected to immunoprecipitation with anti-human Aβ antibody 4G8, and then to western blot using anti-human Aβ antibody 82E1 (Immuno-Biological Laboratories Co., Ltd.). The indicated amounts of human Aβ1-40 (Peptide Institute) were used for the quantification. Data were analyzed using Mann–Whitney non-parametric statistics to assess significance. All animal experiments were approved by the Animal Ethics Committee of Doshisha University prior to their commencement and performed in accordance with approved protocols.

**Statistics and reproducibility**. Significant differences between each group and the control were analyzed using one-way analysis of variance (ANOVA), followed by Dunnett's test. The non-parametric Mann–Whitney *U* test was also used to analyze significant differences in data distribution between groups. In all cases, *P* < 0.05 was considered statistically significant. All statistical analysis was performed using IBM SPSS Statistics software (ver. 26.0.0.0). No statistical methods were used to determine the sample size. We repeated each experiment at least three times and confirmed the reproducibility of each result.

**Reporting summary**. Further information on research design is available in the Nature Portfolio Reporting Summary linked to this article.

## Data availability

All source data presented in the main figures and supplementary figures are available in Supplementary Data 1. All other data or sources are available from the corresponding authors on reasonable request.

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

## Acknowledgements
This work was supported by a grant from the Harris Science Research Institute, Doshisha University; a grant from Japan Science and Technology Agency (JST), Support for Pioneering Research Initiated by the Next Generation (SPRING) Grant Number JPMJSP2129; the Strategic Research Program for Brain Science from the Japan Agency for Medical Research and Development (AMED), Grant Number 19dm0107128h00040 (19dm0107128h0004); Terumo Life Science Foundation; ACRO Research Grants of Teikyo University; and a grant from the Japan Society for the Promotion of Science (JSPS), Grants-in-Aid for Scientific Research (KAKENHI) Grant Number 21K06498.

## Author contributions
W.S., M.W.-T., S.F. and K.N. performed the biological experiments, analyzed and interpreted the data, and wrote the manuscript. T.M and N.U.-T. performed the biophysical experiments, analyzed and interpreted the data, and wrote the manuscript. J.M., M.A., N.N., H.U. and J.T. performed the animal experiments. S.I. performed the biological experiments. M.W.-T. and S.N. synthesized the peptides. K.K., T.S., T.C.-S., S.F. and K.N. supervised the project.

## Competing interests
The authors declare no competing interests.
