## [Peer Review File · Communications Biology]

Reviewers' comments:

Reviewer #1 (Remarks to the Author):

In this work, the authors described the development of a tailored tetravalent peptide, which can inhibit both A β production and aggregation. It is an interesting work with certain novelty. However, several issues need to be addressed before publication.

1. Strategies to reduce A β production directly affect the physiological function of A β monomers, and the authors may need to provide a detailed background in the "Introduction Section" or "Experiment Section" to illustrate the advantages and safety of this therapeutic strategy.
2. It is often not enough to obtain a result by a characterization method alone. For example, the authors indicated that "However, we note that after incubation with LME-tet for 48 h, the total amount of extracellular A β is about 10 times higher than that of intracellular A β , indicating that LME-tet substantially inhibits production of A β in cells". How did the authors come to this conclusion?
3. In animal experiments, the authors mentioned that the reduction of A β plaques observed in the brain was due to the reduction of A β production. This evidence was far from enough, and it might be due to the inhibition of A β aggregation.
4. The authors were advised to distinguish between the inhibition of A β aggregation and the inhibition of A β production.

Reviewer #2 (Remarks to the Author):

Manuscript ID: COMMSBIO-22-2389-T

Title: A tailored tetravalent peptide displays dual functions to inhibit amyloid β production and aggregation

In this manuscript, the authors have used multivalent peptide library screens, and identified a tetravalent peptide, LME-tet, which binds APP and A β via multivalent interactions. They have performed different assays and established that the tetravalent peptide acts as dual functionality to inhibit amyloid β production as well as aggregation. Although the work is well designed and experiments are well organized, however, few concerns have arisen on reading the manuscript which should be addressed.

1. DLS analysis could be useful in monitoring the change in size of the oligomers in absence and presence of LME-tet. It is strongly recommended to incorporate this experiment in the revised version.
2. No thermodynamics has been reported though this would have been an interesting part of the study since the types of interactions involved in the process could have been obtained. The authors could then see what type of interaction occurs between A β and LME-tet. This will further provide a mechanistic understanding of the interaction of A β and LME-tet. The authors should comment on this.
3. For completeness purposes, please include a comparison of the results reported in this work and others.
4. Please include the following references in the introduction part:
[Biomaterials Science, 2022, 10, 3466-3479], [Biomacromolecules, 2021, 22 (11), 4833-4845], [ACS Chemical Neuroscience, 2021, 12 (10), 1737-1748], [ACS Applied Bio Materials, 2020, 3 (10), 6598-6625], [ACS Applied Bio Materials, 2020, 3 (8), 5407-5419]
5. All the references are not in the same format. There are few references where the page number range is missing. Please check this and maintain the reference format same throughout the whole manuscript.

Reviewer #3 (Remarks to the Author):

The manuscript, COMMSBIO-22-2389-T: A tailored tetravalent peptide displays dual functions to inhibit amyloid beta production and aggregation, by Sato W et al investigates the potency of several multivalent peptides to target the Amyloid Precursor Protein (APP), the enzymatic cleavage of which produced Abeta peptides (Ab40 or Ab42) which are prone to amyloid fibrillation and have been historically implicated in Alzheimer's Disease (AD) pathogenesis. The topic of study is of great interest to a general audience as well as to those who support or believe in the idea of amyloid beta-mediated neuronal dysfunction in AD. Despite reports on several possible small molecule inhibitors/drugs that may hinder Ab42 production or extracellular accumulation as larger aggregates (plaques), none have come to practice at clinical levels. Some compounds/peptides have issues crossing the blood-brain barrier; while others show off-target effects or their mechanism of action is poorly understood. In the current work, the authors have followed a systematic approach to select the peptide(s) that would hit the target of choice. They have finally come up with a peptide LME-tet that i) interacts with APP, ii) prevents beta-secretase mediated cleavage of APP, iii) binds to Ab42 and affects its structure and fibrillation kinetics. Finally, LME-tet shows promise in lowering Abeta in an APP expressing mouse model. This work is very nicely designed and executed. While the reviewer appreciates the authors for carrying out an elaborate study and values the outcome of the study for it has crucial clinical significance, there are several areas in the manuscript that required significant changes and improvements. Following are the reviewer's comments and questions for the authors.

1) There is no clear evidence from this work if Abeta production is indeed inhibited. The authors should either devise approaches to study this or alter their claims regarding Abeta production. For instance, Lines 127-133: "...However, we note that after incubation with LME-tet for 48 h, the total amount of extracellular A β is about 10 times higher than that of intracellular A β , indicating that LME-tet substantially inhibits the production of Abeta in cells."

Which figure(s) in the manuscript suggests this? Is it not true that the intracellular and extracellular pools of Abeta are related to each other? Is it possible that a 48-hour treatment showing a ten times higher extracellular Abeta (with LME-tet) than that of intracellular Abeta could simply be an enhanced secretion rather than a decreased production? This idea of Abeta production-to-secretion relation is not well described in the manuscript.

Figure 2c only tells there is decreased extracellular Abeta. How does this data suggest decreased production of Abeta, as the authors claim? In all the figures showing extracellular Abeta, the inclusion of intracellular Abeta data will be useful.

2) What is the purpose of showing 0, 15, 30, and 60 mins immunofluorescence data in Figure 3b, right panel? The 0 and 60 mins data look very similar with respect to the colocalization of APP and the tetravalent peptide. If the authors really wish to see APP dynamics in cells upon LME-tet treatment, a live cell imaging using fluorescent-tagged APP would be the best choice, where the same cell(s) is monitored over time. What does APP immunoblotting data look like on a time scale (0-60 mins) shown in Fig 3b right panel? Do the intracellular APP levels drop or increase with time upon exposure of LME-tet on cells for 60 mins or longer?

3) LysoTracker is generally used for live cell imaging. A little success with LysoTracker staining may be achieved if the fixation is done after staining. Given the poor localization with LAMP1 or LysoTracker with APP in cells, the LysoTracker data may be omitted from this manuscript. However, in case the authors decide to perform a live cell imaging in response to the reviewer's previous comment, studying a possible lysosomal degradation of APP (fluorescent-tagged) may be done using LysoTracker labeling along.

4) The data shown in Figure 4c left panel (St6gal1) is not clear, especially in the last lane which has a beta-secretase inhibitor treated sample. Please replace this with a better representative result.

5) Please include APP-FLAG immunoblotting results in Figure 4d.

6) LME-tet interaction with Ab42 results in fibrils that are shorter. Are these fibrils more prone to entering the neuronal cells? Did the authors test membrane interactions or cytotoxic potential of Ab42 fibrils made in the presence or absence of LME-tet? Given that the on-pathway prefibrillar aggregates

(or oligomers), as well as those released from intact fibrils (upon disassembly), can be toxic to cells, it would be worth studying the said properties of Ab42 fibrils formed in the absence or presence of LME-tet. Also, it would be interesting to see what happens if LME-tet is incubated with preformed Ab42 fibrils. Would this interaction cause dissolution of the fibrils or convert them to a non-fibrillar/amorphous state? Answering this latter question could be reminiscent of a situation in AD where the effect of a drug on the clearance of preexisting Abeta plaques is being studied.

7) Figure 6 shows Abeta levels quantified by immunoblotting. This data is important from a physiological standpoint, therefore has to be validated using additional techniques. Also, the results presented in Figure 6 tell about the peptides amounts/levels, which may or may not necessarily correlate with actual accumulation on the brain tissues unless an accumulation is explicitly shown. One possible approach the authors could attempt here would be to stain the brain tissues (from controls and LMEtet-treated mice) with Congo Red/Thio S and look for colocalization with Abeta-specific antibody staining. Please see the double staining approach shown in this article for your reference: *Am J Pathol.* 2004 Sep; 165(3): 987–995.

8) Can the presence of LME-tet be examined in the brain tissues from the AD model treated with LME-tet?

9) As LME-tet specifically inhibits beta-cleavage of APP, what is the fate of the uncleaved APP in the cells? What degradation pathways in the cells are responsible for the clearance of such species if it is not via lysosomes or autophagosomes?

10) What is the explanation for the increased binding of LME-tet to Ab42 compared to Ab40? Discuss briefly.

11) All graphs should display error bars based on standard deviation (SD) instead of the standard error of the mean (SEM).

12) Please perform a thorough spelling check, and ensure the correctness of references.

Response to comments from Reviewer #1:

We would like to thank Reviewer #1 for considering our manuscript and offering constructive comments that helped to strengthen our conclusions. We have provided point-by-point responses to the Reviewer's concerns below.

Reviewer #1 (Remarks to the Author):

In this work, the authors described the development of a tailored tetravalent peptide, which can inhibit both A β production and aggregation. It is an interesting work with certain novelty.

However, several issues need to be addressed before publication.

1. Strategies to reduce A β production directly affect the physiological function of A β monomers, and the authors may need to provide a detailed background in the "Introduction Section" or "Experiment Section" to illustrate the advantages and safety of this therapeutic strategy.

Response: As the Reviewer noted, the effect of A β reduction on its physiological function must be considered, although its precise physiological function is not fully understood. Peripheral A β was recently shown to be produced by adipose tissues, skeletal muscles, and the liver and function as a negative modulator of insulin secretion (Proc. Natl. Acad. Sci. USA, 2022, 119(12), e2117723119). As shown in the original manuscript, the amount of A β in the cortex and olfactory bulb was substantially reduced by Ac-LME-tet treatment (Fig. 6), whereas the amount of A β in the plasma was not affected (newly added Supplementary Fig. 11), suggesting that Ac-LME-tet does not affect the physiological function of peripheral A β as a regulator of insulin secretion. According to the Reviewer's comment, we added data to the revised version of the manuscript to further emphasize the advantage of this strategy. Accordingly, we added the following sentences:

"On the other hand, the amount of A β in the plasma was not affected (Supplementary Fig. 11). Recently, peripheral A β was shown to be produced by adipose tissues, skeletal muscles, and the liver and function as a negative modulator of insulin secretion (Proc. Natl. Acad. Sci. USA, 2022, 119(12), e2117723119). Thus, our results indicate that Ac-LME-tet does not affect the physiological function of peripheral A β as a regulator of insulin secretion, further emphasizing the advantage of Ac-LME-tet for use as an effective therapeutic compound for AD treatment."
(lines 301–307 in the revised version of the manuscript)

2. It is often not enough to obtain a result by a characterization method alone. For example, the authors indicated that “However, we note that after incubation with LME-tet for 48 h, the total amount of extracellular A β is about 10 times higher than that of intracellular A β , indicating that LME-tet substantially inhibits production of A β in cells”. How did the authors come to this conclusion?

Response: According to the Reviewer’s suggestion, we precisely measured the relative amounts of extracellular and intracellular A β using 7WD10 cells after 48 h of incubation. The amount of extracellular A β was 30.2 times higher than that of intracellular A β , indicating that the decrease in the amount of extracellular A β caused by LME-tet treatment substantially reflects the inhibitory effect of LME-tet on A β production in cells. We added these data to Supplementary Fig. 2 and changed the sentences as follows:

“Among all peptides tested, only LME-tet (containing the Pro-Lys-Leu-Arg-Met-Lys-Glu motif) was able to reduce the amount of extracellular A β to 59% of that in the control, although the intracellular C99 and A β levels were not significantly affected (Fig. 2b). The total amount of extracellular A β was 30.2 ± 4.4 (mean \pm SE; $n = 4$) times higher than that of intracellular A β after incubation of 7WD10 cells for 48 h (Supplementary Fig. 2), indicating that the decrease in the amount of extracellular A β caused by LME-tet treatment substantially reflects the inhibitory effect of LME-tet on A β production in cells.”

(lines 126–133 in the revised version of the manuscript)

3. In animal experiments, the authors mentioned that the reduction of A β plaques observed in the brain was due to the reduction of A β production. This evidence was far from enough, and it might be due to the inhibition of A β aggregation.

Response: In this experiment, Ac-LME-tet was administered to 7-week-old *App*^{NL-G-F/NL-G-F} mice. After 7 days, the amounts of human A β present in the cerebral cortex and olfactory bulb were substantially reduced. At this time, apparent plaque formation, which can be clearly detected after 4–7 months in mouse brains, was not observed as described previously (Nat. Neurosci., 17, 661-, 2014). Thus, it is highly possible that Ac-LME-tet reduces A β production in the brain, but we agree with the Reviewer’s comment that the reduction of the A β level in the brain caused by Ac-LME-tet treatment does not necessarily indicate a reduction of A β production. According to the Reviewer’s suggestion, we changed the following sentence (lines 270–273 in the original version of the manuscript):

“Notably, we detected substantially reduced accumulation of A β in the cortex and the olfactory bulb of mice treated with Ac-LME-tet, as compared to control-treated animals, indicating that Ac-LME-tet can function *in vivo* to reduce production of A β in the brain.”

to

“Notably, we detected substantial reductions in the A β levels in the cortex and olfactory bulb of mice treated with Ac-LME-tet compared with that in control-treated animals, indicating that Ac-LME-tet can function *in vivo* (Fig.6).”

(lines 286–289 in the revised version of the manuscript)

We also changed the following sentences,

“LME-tet efficiently inhibits accumulation of A β ” (line 37)

“LME-tet inhibits accumulation of A β ” (line 261)

“peptide can efficiently inhibit A β accumulation” (line 366)

“Ac-LME-tet inhibits accumulation of A β ” (line 880)

to

“LME-tet efficiently reduces A β levels” (line 37)

“LME-tet reduces A β levels” (line 277)

“peptide can efficiently reduce A β levels” (line 411)

“Ac-LME-tet reduces A β levels” (line 968)

, respectively.

4. The authors were advised to distinguish between the inhibition of A β aggregation and the inhibition of A β production.

Response: According to the Reviewer’s suggestion, we changed the sentence (lines 270–273 in the original version of the manuscript) as described above to clarify this point.

Response to comments from Reviewer #2:

We would like to thank Reviewer #2 for considering our manuscript and offering constructive comments that helped strengthen our conclusions. We have provided point-by-point responses to the Reviewer’s concerns below.

Reviewer #2 (Remarks to the Author):

In this manuscript, the authors have used multivalent peptide library screens, and identified a tetravalent peptide, LME-tet, which binds APP and A β via multivalent interactions. They have performed different assays and established that the tetravalent peptide acts as dual functionality to inhibit amyloid β production as well as aggregation. Although the work is well designed and experiments are well organized, however, few concerns have arisen on reading the manuscript which should be addressed.

1. DLS analysis could be useful in monitoring the change in size of the oligomers in absence and presence of LME-tet. It is strongly recommended to incorporate this experiment in the revised version.

Response: As the Reviewer noted, DLS analysis could be useful to monitor the change in sizes of globular particles in general. However, as shown in Fig. 5d in the original manuscript, after 72 h of incubation, tangled fibril formation and subsequent aggregation of A β 42 were observed in the absence of LME-tet. To respond properly to the Reviewer's request, we performed additional experiments to monitor changes in oligomer/fibril formation of A β 42 at various time points using electron microscopy. As shown in Supplemental Fig. 4 in the revised version of the manuscript, clear fibril formation was detected, even after 4 h of incubation, in the absence of LME-tet. After 8 h of incubation, tangled fibrils and subsequent A β 42 aggregation were observed. LME-tet almost completely inhibited fibril formation for up to 24 h of incubation, which is consistent with the Thioflavin T binding assay results (Fig. 5c in the original version of the manuscript). We added these data to Supplementary Fig. 4 in the revised version of the manuscript and added the following sentences:

“Furthermore, the effect of LME-tet on fibril formation was examined at various time points by electron microscopy (Supplementary Fig. 4). In the absence of LME-tet, fibril formation was detected after 4 h of incubation; additionally, tangled fibrils and subsequent aggregation of A β 42 were observed after 8 h of incubation. LME-tet almost completely inhibited fibril formation for up to 24 h of incubation, which is consistent with the Thioflavin T-binding assay results.”

(lines 250–256 in the revised version of the manuscript)

In addition, we changed the words “oligomer” and “oligomerization” to “fibril” and “fibrillation” in the revised version of the manuscript (lines 257, 262, 263, 597, 598, 601, 946, 957, and 964) because, even after 4 h of incubation, clear fibril formation was detected in the absence of LME-tet as described above.

2. No thermodynamics has been reported though this would have been an interesting part of the study since the types of interactions involved in the process could have been obtained. The authors could then see what type of interaction occurs between A β and LME-tet. This will further provide a mechanistic understanding of the interaction of A β and LME-tet. The authors should comment on this.

Response: As the Reviewer noted, thermodynamic analysis may provide useful information about the types of interactions between A β and LME-tet to show how LME-tet inhibits A β fibrillation. However, in this study, we focused on the finding that LME-tet can potentially inhibit the oligomerization/fibrillation of A β 42 rather than the precise molecular mechanism of the interaction, although we could show some information regarding the mechanism of action by which LME-tet inhibits fibril formation by performing additional electron microscopic analysis as shown above. The data showed that LME-tet clearly inhibited fibril formation even after 4 h of incubation. Combined with the decreased velocity of the conformational change of A β 42 induced by LME-tet (original Fig. 5b), these results demonstrate that LME-tet functions at an early stage to inhibit oligomerization and subsequent pre-fibril formation. To address the Reviewer's comment, we added the following sentences:

“In addition, electron microscopic analysis showed that LME-tet clearly inhibited fibril formation even after 4 h of incubation, further supporting the contention that LME-tet may function at an early stage of pre-fibril formation (Supplementary Fig. 4).”

(lines 393–396 in the revised version of the manuscript)

3. For completeness purposes, please include a comparison of the results reported in this work and others.

Response: According to the Reviewer's comment, we added results for the use of other peptide-based aggregation inhibitors and compared their inhibitory effects on the fibrillation of A β 42 as follows:

“Previous studies showed that a series of monomeric peptides containing a cationic Arg-cluster or amyloidogenic sequences derived from A β , such as the KLVFF (A β 16–20), GGVVIA (A β 37–42), and LPFFD (modified A β 17–21) motifs, act as aggregation inhibitors against A β 42^{49,50, 51,52}. The recently developed and most effective peptide with a modified KLVFF motif, Ac-KL-NVF-NFA-NH₂, which is a diaminated derivative, inhibited fibrillation of A β 42 by 40% with an equivalent molar concentration after 8 h of incubation as measured by the Thioflavin T-binding assay. Additionally, LME-tet completely inhibited fibrillation, even with 1/10 of the molar concentration of A β 42, after 8 h of incubation (Fig. 5c)⁵³.”

(lines 396–404 in the revised version of the manuscript)

4. Please include the following references in the introduction part: [Biomaterials Science, 2022, 10, 3466-3479], [Biomacromolecules, 2021, 22 (11), 4833-4845], [ACS Chemical Neuroscience, 2021, 12 (10), 1737-1748], [ACS Applied Bio Materials, 2020, 3 (10), 6598-6625], [ACS Applied Bio Materials, 2020, 3 (8), 5407-5419]

Response: According to the Reviewer's suggestion, we referenced the important review article (ACS Chemical Neuroscience, 2021, 12 (10), 1737-1748) describing a variety of peptides that inhibit A β 242 fibrillation in the Discussion. However, the other articles, especially those focusing on synthetic polymeric materials, were not necessarily relevant to the present study.

5. All the references are not in the same format. There are few references where the page number range is missing. Please check this and maintain the reference format same throughout the whole manuscript.

Response: We sincerely apologize for the inconvenience. According to the Reviewer's suggestion, we reformatted the references in the revised version of the manuscript as follows:

1 Masters, C. L. *et al.* Amyloid plaque core protein in Alzheimer disease and Down syndrome. *Proc. Natl. Acad. Sci. USA* **82**, 4245-4249 (1985).

10 Novak, G. *et al.* Long-term safety and tolerability of atabecestat (JNJ-54861911), an oral BACE1 inhibitor, in early Alzheimer's disease spectrum patients: a randomized, double-blind, placebo-controlled study and a two-period extension study. *Alzheimer's. Res. Ther.* **12**, Article number: 58 (2020).

12 Rajapaksha, T. W., Eimer, W. A., Bozza, T. C. & Vassar, R. The Alzheimer's beta-secretase enzyme BACE1 is required for accurate axon guidance of olfactory sensory neurons and normal glomerulus formation in the olfactory bulb. *Mol. Neurodegener.* **6**, Article number: 88 (2011).

13 Hitt, B. D., Jaramillo, T. C., Chetkovich, D. M. & Vassar, R. BACE1^{-/-} mice exhibit seizure activity that does not correlate with sodium channel level or axonal localization. *Mol. Neurodegener.* **5**, Article number: 31 (2010).

27 Omi, J. *et al.* The inducible amphisome isolates viral hemagglutinin and defends against influenza A virus infection. *Nat. Commun.* **11**, Article number: 162 (2020).

30 Kitazume, S. *et al.* Alzheimer's β -secretase, β -site amyloid precursor protein-cleaving enzyme, is responsible for cleavage secretion of a Golgi-resident sialyltransferase. *Proc. Natl. Acad. Sci. USA* **98**, 13554-13559 (2001).

- 35 Nishikawa, K. *et al.* A therapeutic agent with oriented carbohydrates for treatment of infections by Shiga toxin-producing *Escherichia coli* O157:H7. *Proc. Natl. Acad. Sci. USA* **99**, 7669-7674 (2002).
- 41 Ma, H. *et al.* Involvement of beta-site APP cleaving enzyme 1 (BACE1) in amyloid precursor protein-mediated enhancement of memory and activity-dependent synaptic plasticity. *Proc. Natl. Acad. Sci. USA* **104**, 8167-8172 (2007).
- 47 Funamoto, S. *et al.* Substrate ectodomain is critical for substrate preference and inhibition of gamma-secretase. *Nat. Commun.* **4**, Article number: 2529 (2013).
- 49 Mamsa, S. S. A. & Meloni, B. P. Arginine and arginine-rich peptides as modulators of protein aggregation and cytotoxicity associated with Alzheimer's disease. *Front. Mol. Neurosci.* **14**, Article number: 759729 (2021).
- 56 Watanabe-Takahashi, M. *et al.* Identification of a peptide motif that potently inhibits two functionally distinct subunits of Shiga toxin. *Commun Biol.* **4**, Article number: 538 (2021).

Response to comments from Reviewer #3:

We would like to thank Reviewer #3 for considering our manuscript and offering constructive comments that helped to strengthen our conclusions. We have provided point-by-point responses to the Reviewer's concerns below.

Reviewer #3 (Remarks to the Author):

The manuscript, COMMSBIO-22-2389-T: A tailored tetravalent peptide displays dual functions to inhibit amyloid beta production and aggregation, by Sato W et al investigates the potency of several multivalent peptides to target the Amyloid Precursor Protein (APP), the enzymatic cleavage of which produced Abeta peptides (Ab40 or Ab42) which are prone to amyloid fibrillation and have been historically implicated in Alzheimer's Disease (AD) pathogenesis. The topic of study is of great interest to a general audience as well as to those who support or believe in the idea of amyloid beta-mediated neuronal dysfunction in AD. Despite reports on several possible small molecule inhibitors/drugs that may hinder Ab42 production or extracellular accumulation as larger aggregates (plaques), none have come to practice at clinical levels. Some compounds/peptides have issues crossing the blood-brain barrier; while others show off-target effects or their mechanism of action is poorly understood. In the current work, the authors have followed a systematic approach to select the peptide(s) that would hit the target of choice. They have finally come up with a peptide LME-tet that i) interacts with APP, ii)

prevents beta-secretase mediated cleavage of APP, iii) binds to Ab42 and affects its structure and fibrillation kinetics. Finally, LME-tet shows promise in lowering Abeta in an APP expressing mouse model. This work is very nicely designed and executed. While the reviewer appreciates the authors for carrying out an elaborate study and values the outcome of the study for it has crucial clinical significance, there are several areas in the manuscript that required significant changes and improvements. Following are the reviewer's comments and questions for the authors.

1) There is no clear evidence from this work if Abeta production is indeed inhibited. The authors should either devise approaches to study this or alter their claims regarding Abeta production.

For instance, Lines 127-133: "...However, we note that after incubation with LME-tet for 48 h, the total amount of extracellular A β is about 10 times higher than that of intracellular A β , indicating that LME-tet substantially inhibits the production of Abeta in cells."

Which figure(s) in the manuscript suggests this? Is it not true that the intracellular and extracellular pools of Abeta are related to each other? Is it possible that a 48-hour treatment showing a ten times higher extracellular Abeta (with LME-tet) than that of intracellular Abeta could simply be an enhanced secretion rather than a decreased production? This idea of Abeta production-to-secretion relation is not well described in the manuscript.

Figure 2c only tells there is decreased extracellular Abeta. How does this data suggest decreased production of Abeta, as the authors claim? In all the figures showing extracellular Abeta, the inclusion of intracellular Abeta data will be useful.

Response: According to the Reviewer's suggestion, we precisely measured the relative amounts of extracellular and intracellular A β using 7WD10 cells after 48 h of incubation. The amount of extracellular A β was 30.2 times higher than that of intracellular A β , indicating that the decrease in the extracellular A β amount caused by LME-tet treatment substantially reflects the inhibitory effect of LME-tet on A β production in cells. We added the data to Supplementary Fig. 2 in the revised version of the manuscript and changed the sentences as follows:

"Among all peptides tested, only LME-tet (containing the Pro-Lys-Leu-Arg-Met-Lys-Glu motif) was able to reduce the amount of extracellular A β to 59% of that in the control, although the intracellular C99 and A β levels were not significantly affected (Fig. 2b). The total amount of extracellular A β was 30.2 ± 4.4 (mean \pm SE; $n = 4$) times higher than that of intracellular A β after incubation of 7WD10 cells for 48 h (Supplementary Fig. 2), indicating that the decrease in the amount of extracellular A β by LME-tet treatment substantially reflects the inhibitory effect of LME-tet on A β production in cells."

(lines 126–133) in the revised version of the manuscript)

2) What is the purpose of showing 0, 15, 30, and 60 mins immunofluorescence data in Figure 3b, right panel? The 0 and 60 mins data look very similar with respect to the colocalization of APP and the tetravalent peptide. If the authors really wish to see APP dynamics in cells upon LME-tet treatment, a live cell imaging using fluorescent- tagged APP would be the best choice, where the same cell(s) is monitored over time. What does APP immunoblotting data look like on a time scale (0-60 mins) shown in Fig 3b right panel? Do the intracellular APP levels drop or increase with time upon exposure of LME-tet on cells for 60 mins or longer?

Response: The purpose of Fig. 3b (right panel) is to show that LME-tet initially binds to the APP present in the plasma membrane and is endocytosed into endosomes in a complex with APP. In the original Fig. 3b (right panel), after 60 min of incubation, colocalization of LME-tet with APP could be detected as puncta present in the cytosol but not in the plasma membrane, which is difficult to see as the Reviewer notes. To clarify this point, we replaced the image with a new representative image as shown in the revised Fig. 3b (right panel).

In Fig. 3b, cells were treated with the anti-APP antibody 6E10 on ice for 30 min to label the APP present in the plasma membrane and then incubated at 37°C for the indicated time. This procedure enabled us to monitor the intracellular transport of APP originally present in the plasma membrane, which cannot be clearly analyzed using fluorescence-tagged APP expressed in the cells.

LME-tet treatment at 37°C for 30 min did not largely affect the total amount of APP present in the cells as shown in Fig. 3a (lower panel; Input). However, after 48 h of treatment with LME-tet, the total amount of APP (mature and immature form) increased, consistent with the reduction of the amount of extracellular A β (Fig. 2b, 2c, and 3d).

3) LysoTracker is generally used for live cell imaging. A little success with LysoTracker staining may be achieved if the fixation is done after staining. Given the poor localization with LAMP1 or LysoTracker with APP in cells, the LysoTracker data may be omitted from this manuscript. However, in case the authors decide to perform a live cell imaging in response to the reviewer's previous comment, studying a possible lysosomal degradation of APP (fluorescent-tagged) may be done using LysoTracker labeling along.

Response: Based on the Reviewer's comment, we calculated Pearson's coefficient to evaluate the difference in colocalization between APP and LysoTracker in the presence or absence of LME-tet. This analysis can provide a quantitative evaluation when only faint colocalization is

observed. As shown in the revised version of Fig. 3c (right panel), no significant difference was observed. Accordingly, we added the following sentences to the legend of Fig. 3c:

“Pearson’s coefficient for colocalization between APP and Lysotracker is shown (right panel; mean \pm SE, n (number of cells) = 10-24 or 7-14, for the absence (-) from five independent experiments or presence of LME-tet from three independent experiments, respectively). Pearson’s coefficient was calculated using the Coloc 2 plugin in Image J. The significance vs. LME-tet (-) was calculated using ANOVA, followed by a one-sided Dunnett’s test. n.s., not significant.”

(lines 903–909 in the revised version of the manuscript)

As described in Fig. 3b and Fig. 3c, cells were treated with the anti-APP antibody 6E10 on ice for 30 min to label APP present in the plasma membrane. Then, the cells were incubated at 37°C for 60 min to examine the colocalization of Lysotracker and endocytosed APP, which cannot be analyzed using expression of fluorescent-tagged APP in the cells.

4) The data shown in Figure 4c left panel (St6gal1) is not clear, especially in the last lane which has a beta- secretase inhibitor treated sample. Please replace this with a better representative result.

Response: According to the Reviewer’s suggestion, we replaced the panel with a representative image as shown in Fig. 4c in the revised version of the manuscript.

5) Please include APP-FLAG immunoblotting results in Figure 4d.

Response: According to the Reviewer’s suggestion, we added a panel showing APP633-685-FLAG immunoblotting as shown in Fig. 4d in the revised version of the manuscript.

6) LME-tet interaction with Ab42 results in fibrils that are shorter. Are these fibrils more prone to entering the neuronal cells? Did the authors test membrane interactions or cytotoxic potential of Ab42 fibrils made in the presence or absence of LME-tet? Given that the on-pathway prefibrillar aggregates (or oligomers), as well as those released from intact fibrils (upon disassembly), can be toxic to cells, it would be worth studying the said properties of Ab42 fibrils formed in the absence or presence of LME-tet. Also, it would be interesting to see what happens if LME-tet is incubated with preformed Ab42 fibrils. Would this interaction cause dissolution of the fibrils or convert them to a non-fibrillar/amorphous state? Answering this

latter question could be reminiscent of a situation in AD where the effect of a drug on the clearance of preexisting Abeta plaques is being studied.

Response: According to the Reviewer's suggestion, we performed additional experiments to examine the cytotoxic potential of A β 42 fibrils or pre-fibrils formed in the absence or presence of Ac-LME-tet using the Neuro-2a neuronal cell line. Incubation of the cells with A β 42 for 48 h in the absence of Ac-LME-tet reduced cell viability by 20%. The presence of Ac-LME-tet rescued the viability by 5% (Supplementary Fig. 7 in the revised version of the manuscript), indicating that Ac-LME-tet can potentially protect cells by retarding the formation of cytotoxic prefibrillar aggregates or oligomers, consistent with the observations shown in Fig. 5 and Supplementary Fig. 4 in the revised version of the manuscript. Accordingly, we added the following sentences:

“Previously, prefibrillar aggregates or oligomers of A β 42, but not fully aggregated fibrils, were shown to have cytotoxic activity against neuronal cells^{32,33}. We examined the protective effect of Ac-LME-tet on the cytotoxic activity of A β 42 prefibrillar aggregates using the Neuro-2a neuronal cell line. Incubation of the cells with A β 42 for 48 h in the absence of Ac-LME-tet reduced cell viability by 20%. The presence of Ac-LME-tet rescued the viability by 5% (Supplementary Fig. 7), indicating that Ac-LME-tet can potentially protect cells by retarding the formation of cytotoxic prefibrillar aggregates or oligomers, consistent with the above observations (Fig. 5 and Supplementary Fig. 4).”

(lines 267–275 in the revised version of the manuscript)

According to the Reviewer's comment, we performed an additional experiment to examine whether LME-tet causes the dissolution of preformed A β 42 fibrils, but LME-tet did not affect the fibril condition (attached Fig. A1). As shown in Fig. 5e, LME-tet could potentially impede the enhanced growth of A β 42 fibrils from preformed A β 42 fibril-seed, but after 24 h of incubation, the level of fibril formation was similar to that observed in the absence of LME-tet. In contrast, as shown in Fig. 5c, in the absence of preformed A β 42 fibril-seed, sufficient inhibition of fibril formation by LME-tet could be observed for up to 96 h of incubation. Thus, LME-tet can inhibit the initial formation of A β 42 fibril-seed, which markedly stimulates the subsequent growth of fibrils, rather than causing dissolution of fibrils. This finding is also supported by the additional electron microscopic analysis data (Supplementary Fig. 4), which shows the highly efficient inhibitory effect of LME-tet on A β 42 fibril-seed formation, even after 24 h of incubation. Accordingly, we added the following sentence:

“Thus, combined with the results of Fig. 5c and Supplementary Fig. 4, LME-tet can inhibit the initial formation of A β 42 fibril-seed, which markedly stimulates the subsequent growth of the fibrils.”

(lines 264–266 in the revised version of the manuscript)

7) Figure 6 shows Abeta levels quantified by immunoblotting. This data is important from a physiological standpoint, therefore has to be validated using additional techniques. Also, the results presented in Figure 6 tell about the peptides amounts/levels, which may or may not necessarily correlate with actual accumulation on the brain tissues unless an accumulation is explicitly shown. One possible approach the authors could attempt here would be to stain the brain tissues (from controls and LMEtet-treated mice) with Congo Red/Thio S and look for colocalization with Abeta-specific antibody staining. Please see the double staining approach shown in this article for your reference: *Am J Pathol.* 2004 Sep; 165(3): 987–995.

Response: We appreciate the Reviewer’s comment. In this experiment, Ac-LME-tet was administered to 7-week-old *App*^{NL-G-F/NL-G-F} mice. After 7 days, the amounts of human A β present in the cerebral cortex and olfactory bulb were substantially reduced. At this time, apparent plaque formation, which can be clearly detected after 4–7 months in the mouse brain, was not observed as described previously (*Nat. Neurosci.*, 17, 661-, 2014). As the Reviewer noted, it is important to examine the effect of LME-tet on A β plaque formation in brain tissues, but in this study, we focused on the inhibitory effect of Ac-LME-tet on initial A β accumulation in the brain.

8) Can the presence of LME-tet be examined in the brain tissues from the AD model treated with LME-tet?

Response: According to the Reviewer’s suggestion, we examined the distribution of FITC-labeled LME-tet in the brain after intraperitoneal administration using *App*^{NL-G-F/NL-G-F} mice. Although no peptide signals were observed in individual neurons, we detected substantial peptide signals in the choroid plexus of the lateral ventricle (CPLV) of the brain, and the images were quite similar to those of peptide #4, which also reduced A β levels both *in vitro* and *in vivo* (as mentioned in lines 376-377 in the revised version of the manuscript). Thus, a certain amount of the peptide may penetrate the blood–brain barrier of the CPLV and enter the brain. In addition, we compared the efficiency of LME-tet incorporation into cells with that of another identified peptide, PME-tet,

which does not inhibit A β production (Fig. 2b). We found that LME-tet can be incorporated with much higher efficiency. This feature of LME-tet, to be easily transported into cells through the cellular membrane, may contribute to its efficient localization in the brain. We added these data as Supplementary Figs. 9 and 10 to the revised version of the manuscript and added the following sentences:

“To investigate whether Ac-LME-tet penetrates the brain, we intraperitoneally administered FITC-labeled LME-tet and examined the brain distribution of the peptide. Although no peptide signals were observed in individual neurons, we detected substantial peptide signals in the choroid plexus of the lateral ventricle (CPLV) of the brain (Supplementary Fig. 9). Thus, a certain amount of the peptide may penetrate the blood–brain barrier of the CPLV and enter the brain. In addition, we compared the efficiency of LME-tet incorporation into cells with that of another identified peptide, PME-tet, which does not inhibit A β production (Fig. 2b). We found that LME-tet can be incorporated with much higher efficiency than PME-tet (Supplementary Fig. 10). This feature of LME-tet, to be easily transported to cells through the cellular membrane, may contribute to its efficient localization in the brain.”

(lines 289-300 in the revised version of the manuscript)

9) As LME-tet specifically inhibits beta-cleavage of APP, what is the fate of the uncleaved APP in the cells? What degradation pathways in the cells are responsible for the clearance of such species if it is not via lysosomes or autophagosomes?

Response: As shown in Figs. 3d and 4a, LME-tet increased the amount but did not induce major degradation of uncleaved APP, even though a small portion of the accumulated APP was degraded through lysosomes or autophagosomes, as observed in the absence of LME-tet (Fig. 3d; PI treatment slightly increased the amount of APP both in the absence and presence of LME-tet). To clarify this point, we changed the following sentence (lines 194-196 in the original manuscript):

“We found that total amounts of glycosylated APP (mature form) and immature form are increased by LME-tet treatment (Fig. 4a).”

to

“We found that the total amounts of glycosylated APP (mature form) and the immature form were increased by 1.6 and 1.3 times, respectively, by LME-tet treatment (Fig. 4a).”

(lines 194-196 in the revised version of the manuscript)

10) What is the explanation for the increased binding of LME-tet to Ab42 compared to Ab40? Discuss briefly.

Response: As shown in lines 335-338 in the revised version of the manuscript, in general, high-avidity intermolecular interactions based on the clustering effect have been observed between multivalent ligands, including multivalent peptides or carbohydrate-polymers, and target molecules prone to cluster or that function via multi-subunit structures. Thus, the higher binding affinity of LME-tet to A β 42 than to A β 40 may reflect the propensity of the former to readily form oligomers, such as dimers, trimers, and multimers, in which each LME-tet binding site is clustered in close proximity to exert a clustering effect on binding. According to the Reviewer's suggestion and to clarify this point, we changed the following sentence (lines 304-306 in the original manuscript):

“Thus, we expect that the relatively higher binding affinity of LME-tet to A β 42 vs. A β 40 may reflect the propensity of the former to readily form a clustered structure.”

to

“Thus, we expect that the higher binding affinity of LME-tet to A β 42 than to A β 40 may reflect the propensity of the former to readily form oligomers, such as dimers, trimers, and multimers, in which each LME-tet binding site is clustered in close proximity to exert a clustering effect on the binding.”

(lines 338-342 in the revised version of the manuscript)

11) All graphs should display error bars based on standard deviation (SD) instead of the standard error of the mean (SEM).

Response: In all graphs in this study, standard errors were used because we wanted to estimate the reliability of the mean but not the variation of each data set.

12) Please perform a thorough spelling check, and ensure the correctness of references.

Response: We sincerely apologize for the inconvenience. According to the Reviewer's suggestion, we performed an additional spelling check and rearranged the references in the revised version of the manuscript as follows:

lines 396-397,

“Previous studies showed that a series of monomeric peptides containing a cationic Arg-cluster or amyloidogenic sequences derived from A β ,”

References

- 1 Masters, C. L. *et al.* Amyloid plaque core protein in Alzheimer disease and Down syndrome. *Proc. Natl. Acad. Sci. USA* **82**, 4245-4249 (1985).
- 10 Novak, G. *et al.* Long-term safety and tolerability of atabecestat (JNJ-54861911), an oral BACE1 inhibitor, in early Alzheimer's disease spectrum patients: a randomized, double-blind, placebo-controlled study and a two-period extension study. *Alzheimer's. Res. Ther.* **12**, Article number: 58 (2020).
- 12 Rajapaksha, T. W., Eimer, W. A., Bozza, T. C. & Vassar, R. The Alzheimer's beta-secretase enzyme BACE1 is required for accurate axon guidance of olfactory sensory neurons and normal glomerulus formation in the olfactory bulb. *Mol. Neurodegener.* **6**, Article number: 88 (2011).
- 13 Hitt, B. D., Jaramillo, T. C., Chetkovich, D. M. & Vassar, R. BACE1-/- mice exhibit seizure activity that does not correlate with sodium channel level or axonal localization. *Mol. Neurodegener.* **5**, Article number: 31 (2010).
- 27 Omi, J. *et al.* The inducible amphisome isolates viral hemagglutinin and defends against influenza A virus infection. *Nat. Commun.* **11**, Article number: 162 (2020).
- 30 Kitazume, S. *et al.* Alzheimer's β -secretase, β -site amyloid precursor protein-cleaving enzyme, is responsible for cleavage secretion of a Golgi-resident sialyltransferase. *Proc. Natl. Acad. Sci. USA* **98**, 13554-13559 (2001).
- 35 Nishikawa, K. *et al.* A therapeutic agent with oriented carbohydrates for treatment of infections by Shiga toxin-producing Escherichia coli O157:H7. *Proc. Natl. Acad. Sci. USA* **99**, 7669-7674 (2002).
- 41 Ma, H. *et al.* Involvement of beta-site APP cleaving enzyme 1 (BACE1) in amyloid precursor protein-mediated enhancement of memory and activity-dependent synaptic plasticity. *Proc. Natl. Acad. Sci. USA* **104**, 8167-8172 (2007).
- 47 Funamoto, S. *et al.* Substrate ectodomain is critical for substrate preference and inhibition of gamma-secretase. *Nat. Commun.* **4**, Article number: 2529 (2013).
- 49 Mamsa, S. S. A. & Meloni, B. P. Arginine and arginine-rich peptides as modulators of protein aggregation and cytotoxicity associated with Alzheimer's disease. *Front. Mol. Neurosci.* **14**, Article number: 759729 (2021).
- 56 Watanabe-Takahashi, M. *et al.* Identification of a peptide motif that potently inhibits two functionally distinct subunits of Shiga toxin. *Commun. Biol.* **4**, Article number: 538 (2021).

Attached Fig. A1

Attached Fig. A1: LME-tet does not degrade A β 42 fibrils.

Preformed A β 42 fibrils were prepared by incubating A β 42 (20 μ M) in 50 mM sodium phosphate buffer (pH 7.4) for 48 h at 37°C. After washing by centrifugation, the fibrils were incubated in the presence or absence of LME-tet (20 μ M) or LME-mono (20 μ M) for the indicated times. Fibrillation profiles of A β 42 were measured using the Thioflavin T-binding assay. Data represent the mean \pm SE; $n = 3$.

REVIEWERS' COMMENTS:

Reviewer #1 (Remarks to the Author):

The authors have addresses most of the concerns. The paper can be published now.

Reviewer #2 (Remarks to the Author):

This paper has been improved significantly, and I recommend it be published in this journal as it is.

Reviewer #3 (Remarks to the Author):

The authors have satisfactorily addressed the reviewer's comments and questions. A final spelling check would be required which may be done either in the time of proofreading by the authors or during typesetting.